# Evolution of Concepts in Language Model Pre-Training

**Xuyang Ge[†], Wentao Shu[†], Jiaxing Wu[†], Yunhua Zhou[‡], Zhengfu He[†], Xipeng Qiu[†] [*]**

[†] OpenMOSS Team, Shanghai Innovation Institute; Fudan University
[‡] Shanghai AI Laboratory
xyge24@m.fudan.edu.cn, zfhe19@fudan.edu.cn

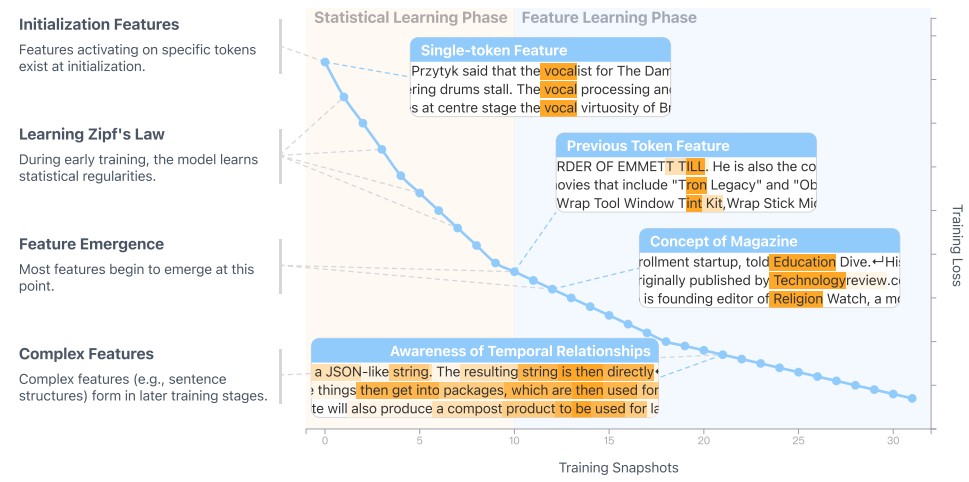

## Abstract

Language models obtain extensive capabilities through pre-training. However, the pre-training dynamics remains a black box. In this work, we track linear interpretable feature evolution across pre-training snapshots using a sparse dictionary learning method called crosscoders. We find that most features begin to form around a specific point, while more complex patterns emerge in later training stages. Feature attribution analyses reveal causal connections between feature evolution and downstream performance. Our feature-level observations are highly consistent with previous findings on Transformer's two-stage learning process, which we term a statistical learning phase and a feature learning phase. Our work opens up the possibility to track fine-grained representation progress during language model learning dynamics. Our code is available at https://github.com/OpenMOSS/Language-Model-SAEs.

## 1 Introduction

Pre-training (Radford et al., 2018; Devlin et al., 2019) has emerged as the dominant paradigm for developing frontier large language models (LLM) (OpenAI, 2023; Grattafiori et al., 2024; Yang et al., 2025). Despite its remarkable success, the pre-training process remains largely a black box. While scaling laws (Hestness et al., 2017; Kaplan et al., 2020; Bahri et al., 2021) reveal the predictable relationships between compute, data, and loss, they offer limited insight into the internal reorganization occurring within the model parameters. Other theoretical frameworks on learning dynamics, including neural tangent kernel (Jacot et al., 2018), information bottleneck theory (Tishby & Zaslavsky, 2015; Shwartz-Ziv & Tishby, 2017), and singular learning theory (Watanabe, 2009; Lau

---

[*]Corresponding author.

et al., 2024; Wang et al., 2025a) — provide high-level explanations on *why* we can observe generalization and grokking (Power et al., 2022; Nanda et al., 2023). However, a fundamental question — *how* do LLMs actually develop their capabilities internally during pre-training — has remained largely opaque.

Recent advances in mechanistic interpretability, particularly through dictionary learning methods based on sparse autoencoders (SAEs), have begun to illuminate the internal representation of neural networks (Bricken et al., 2023; Templeton et al., 2024; Gao et al., 2025). By disentangling the phenomenon of superposition (Elhage et al., 2022), sparse autoencoders show their capabilities in extracting millions of features from LLM activations, demonstrating that LLMs encode human-interpretable concepts as linear directions in their activation spaces. However, these analyses have predominantly focused on studying fully-trained models. Consequently, the process of how features initially emerge and evolve throughout the training process remains largely unexplored.

In this paper, we propose to **track feature evolution** across pre-training snapshots using **cross-coders** (Lindsey et al., 2024), a variant of sparse autoencoders designed to simultaneously identify and align features from a family of correlated model activations. While originally introduced to resolve cross-layer superposition and track features distributed across layers, we adapt this approach to analyze activations from different training checkpoints. By applying cross-snapshot crosscoders, we can observe where features emerge, rotate, and degenerate, thereby providing deeper insight into the internal dynamics of model pre-training. Our main contributions are as follows:

1. To the best of our knowledge, this work is the first to adapt crosscoders to study training dynamics (Section 3). We evaluate completeness and faithfulness of interpretations provided by crosscoders in Appendix A.

2. We perform in-depth analyses on cross-snapshot features in Section 4. We empirically show that the decoder norms can serve as proxies of feature evolution status, and showcase both general and per-feature evolutionary properties.

3. Our method successfully connects the microscopic features to the macroscopic downstream task metrics using attribution-based circuit tracing techniques (Section 5).

4. We show evidence for the phase transition from a statistical learning phase to a feature learning phase in Section 6.

## 2 RELATED WORKS

**Learning dynamics and phase transitions.** Multiple theoretical frameworks explain neural network training dynamics. Neural Tangent Kernel (NTK) theory (Jacot et al., 2018) establishes that infinite-width networks evolve as kernel machines with fixed kernels during gradient descent, with extensions to finite-width corrections (Dyer & Gur-Ari, 2020) and modern architectures (Yang, 2020; Yang & Littwin, 2021). Recent work identifies phase transitions and lazy regimes in training dynamics (Kumar et al., 2024; Zhou et al., 2025). Information Bottleneck (IB) theory (Tishby & Zaslavsky, 2015) formulates deep learning as optimizing compression-prediction tradeoffs. Shwartz-Ziv & Tishby (2017) empirically demonstrates two distinct training phases—initial fitting followed by compression—which aligns with our findings in Section 6. Singular Learning Theory (SLT) treats neural networks as singular statistical models. Watanabe (1999; 2009) introduces the Real Log Canonical Threshold to provide geometric complexity measures predicting phase transitions, with recent advances enabling practical estimation at scale through Local Learning Coefficients (Lau et al., 2024; Furman & Lau, 2024; Wang et al., 2025a). Our work complements these theoretical frameworks by providing detailed mechanistic accounts of feature evolution during transformer pre-training, bridging high-level theory with empirical observations.

**Sparse dictionary learning.** The superposition hypothesis posits that models use linear representations (Bengio et al., 2014; Alain & Bengio, 2017; Vargas & Cotterell, 2020) to embed more features than neurons (Olah et al., 2020; Elhage et al., 2022). Sparse Autoencoders (SAEs) (Bricken et al., 2023; Huben et al., 2024; He et al., 2024) extract monosemantic features from superposition using sparse dictionary learning, with subsequent improvements and scaling efforts (Gao et al., 2025; Rajamanoharan et al., 2024). Ge et al. (2024) and Dunefsky et al. (2024) propose transcoders, an SAE variant that predicts future activations to improve circuit tracability. Recently, Lindsey et al.

(2024) introduces crosscoders to simultaneously read and write to multiple layers. Mishra-Sharma et al. (2024); Minder et al. (2025) leverages crosscoders for capturing the difference between pre-trained models and their chat-tuned versions.

## 3 TRACKING THE EVOLUTION OF FEATURES

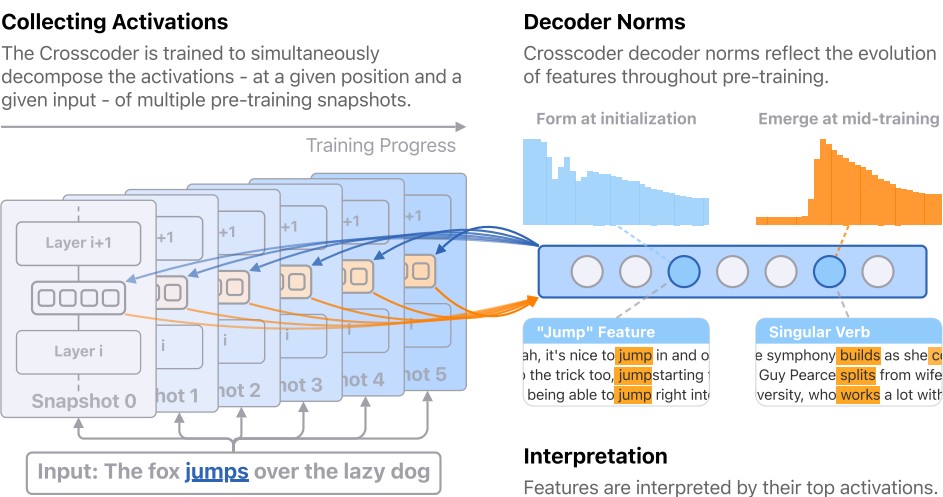

Figure 1: Overview of our method. The crosscoder is trained to decompose activations of multiple pre-training snapshots (left) into sparse features (right).

**Crosscoder architecture.** For a given text corpus $\mathcal{C}$ and a family of training snapshots $\Theta$ saved during LLM pre-training, let $\mathcal{A} = \{a^\theta : \mathcal{C} \times \mathbb{N} \to \mathbb{R}^{d_{\text{model}}} \mid \theta \in \Theta\}$ denote a corresponding family of parameterized functions mapping input datapoints to model activations at a specific layer. Each datapoint $x = (c, j) \in \mathcal{C} \times \mathbb{N}$ is a training token in $\mathcal{C}$, indexing the $j$-th token in sequence $c$.

A cross-snapshot crosscoder (Lindsey et al., 2024) operates on the parameterized function family $\mathcal{A}$ over corpus $\mathcal{C}$ (Figure 1). The crosscoder architecture is defined by:

$$f(x) = \sigma \left( \sum_{\theta \in \Theta} W_{\text{enc}}^\theta a^\theta(x) + b_{\text{enc}} \right)$$
$$\hat{a}^\theta(x) = W_{\text{dec}}^\theta f(x) + b_{\text{dec}}^\theta \tag{1}$$

with $W_{\text{enc}}^\theta \in \mathbb{R}^{n_{\text{features}} \times d_{\text{model}}}$, $b_{\text{enc}} \in \mathbb{R}^{n_{\text{features}}}$, $W_{\text{dec}}^\theta \in \mathbb{R}^{d_{\text{model}} \times n_{\text{features}}}$, and $b_{\text{dec}}^\theta \in \mathbb{R}^{d_{\text{model}}}$. The parameters $W_{\text{enc}}^\theta$, $W_{\text{dec}}^\theta$, and $b_{\text{dec}}^\theta$ correspond to snapshot-specific encoder and decoder weights for parameter $\theta$. The activation function $\sigma(\cdot)$ produces sparse feature activations $f(x)$ shared for all snapshots, and $\hat{a}^\theta(x)$ denotes the reconstructed term of model activation $a^\theta(x)$.

**Training objectives.** The crosscoder is trained to minimize the loss:

$$\mathcal{L}(x) = \underbrace{\sum_{\theta \in \Theta} ||a^\theta(x) - \hat{a}^\theta(x)||^2}_{\text{Reconstruction loss}} + \lambda_{\text{sparsity}} \underbrace{\sum_{\theta \in \Theta} \sum_{i=1}^{n_{\text{features}}} \Omega \left( f_i(x) \cdot ||W_{\text{dec},i}^\theta|| \right)}_{\text{Sparsity loss}} \tag{2}$$

where $\lambda_{\text{sparsity}}$ is a hyperparameter to control the trade-off between reconstruction fidelity and feature sparsity. The regularization function $\Omega(\cdot)$ serves as a differentiable substitute for L0 regularization, penalizing non-sparse feature activations. We include the decoder norm $||W_{\text{dec},i}^\theta||$ in the regularization term to prevent the crosscoder from trivially reducing the feature activation $f_i(x)$ while inflating the decoder norms under imperfect L0 approximations such as L1 regularization. Appendix A.1

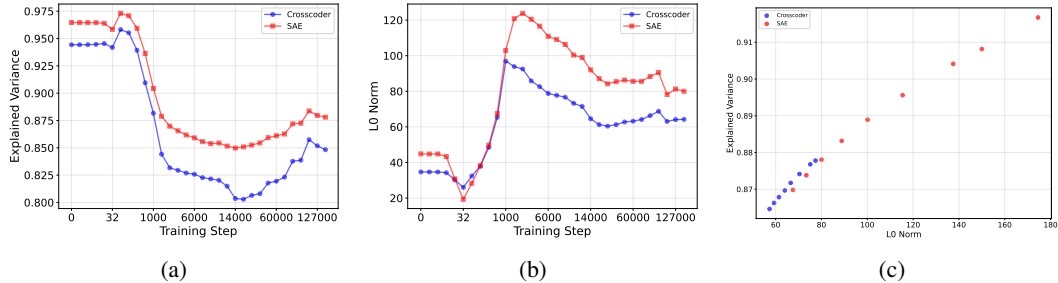

Figure 3: Comparison between crosscoders and per-snapshot SAEs. (a) The explained variance of crosscoders versus SAEs at each snapshot. (b) The L0 norm of crosscoders versus SAEs at each snapshot. (c) The Pareto frontier comparison of crosscoders and SAEs trained on the final snapshot.

shows the details for selecting the proper activation function $\sigma(\cdot)$ and regularization function $\Omega(\cdot)$ to optimize crosscoder feature sparsity.

**Experimental setup.** We use Pythia-160M and Pythia-6.9B (Biderman et al., 2023) for our experiments throughout this work. Pythia is a Transformer language model suite with well-controlled training settings and accessibility to training snapshots. We select the middle layers (Layer 6 of Pythia-160M and Layer 16 of Pythia-6.9B) for training crosscoders. To balance training cost with the granularity of feature evolution analysis, we select 32 snapshots out of 154 open-source snapshots ranging from step 0 to 143,000, with a stratified sampling approach: (1) all 20 snapshots before step 10,000 to capture early feature evolution with maximum temporal resolution, and (2) 12 evenly spaced snapshots from later training stages, containing 4 snapshots from steps 14,000–34,000 and 8 snapshots from steps 47,000–143,000. We use SlimPajama (Shen et al., 2023), a comprehensive text dataset covering a variety of data sources to sample activations.

**Results.** Figure 2 demonstrates crosscoder performance in decomposing model activations into sparse feature representations. We evaluate reconstruction quality (measured by activation variance explained) and sparsity (measured by L0 norm averaged across snapshots). Increasing dictionary size $n_{\text{features}}$ yields significant Pareto improvements across both metrics.

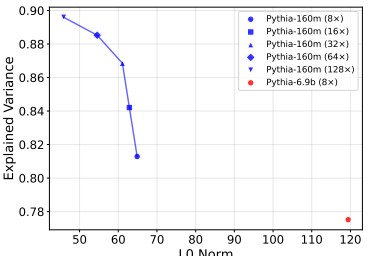

Figure 2: Explained variances versus L0 norms of our crosscoders.

To examine whether crosscoders can extract cross-snapshot features and align identical features in consistent directions within the feature space, we compare crosscoder performance against corresponding SAEs trained individually on each Pythia snapshot. We train SAEs using identical settings and hyperparameters as crosscoders on each snapshot. Figures 9a and 9b demonstrate that crosscoders achieve comparable performance in L0 norm and explained variance at each snapshot, even when SAEs are optimized for individual snapshots where they should have a theoretical advantage.

We further compare the Pareto frontiers (explained variance versus L0 norm) between crosscoders and SAEs trained on the final snapshot. Figure 9c shows that crosscoders exhibit a slightly superior Pareto frontier, demonstrating their effectiveness in sparse dictionary learning beyond their primary capability of tracking feature evolution across snapshots.

**Feature evolution revealed by crosscoders.** An important advantage of crosscoders is the unified feature space (or *sparse codes*) they reveal. The crosscoder encoder aggregates cross-snapshot information to produce shared feature activations. Then these activations are translated back to recover the original activations by a group of independent decoders (or *dictionaries*).

If a feature activates but "exists" at only a subset of snapshots, the sparse penalty will suppress the decoder norms of this feature at irrelevant snapshots to near-zero so they won't interfere with reconstruction on these snapshots and also reduce sparsity loss.

This design principle leads to a crucial observation: the decoder norm $\|W_{\text{dec},i}^{\theta}\|$ directly reflects the strength and presence of feature $i$ at snapshot $\theta$. Therefore, tracking decoder norm changes across snapshots provides a direct window into feature evolution dynamics. Appendix C provides a sanity check of whether $\|W_{\text{dec},i}^{\theta}\|$ can indeed serve as a proxy of feature strength using linear probes.

**Potential failure modes of crosscoder feature alignment.** Before assessing crosscoder features, we further discuss two theoretical concerns about cross-snapshot feature alignment:

*Will crosscoders misalign unrelated features?* Such misalignment would strongly contradict the optimization objective of crosscoders. The distinct activation patterns of unrelated features would conflict when forced into the same dimension. Once a feature activates, it would cause others to activate simultaneously, introducing noise in the reconstruction. In terms of results, misalignment would lead to the polysemanticity of crosscoder features, damaging the consistency of their interpretation, which our interpretability evaluation demonstrates does not occur.

*Will crosscoders split shared features?* Suppose features from different snapshots represent the same underlying concept with identical activation patterns, splitting them across multiple dimensions would be suboptimal: it wastes representational capacity in feature space, as both previous works (Gao et al., 2025; Templeton et al., 2024) and Appendix A.4 prove that feature space dimensionality is crucial for better reconstruction fidelity. Nevertheless, in rare cases, we do observe feature splitting among highly active features. These features share semantic and directional similarity but exhibit subtly different activation patterns that justify separate dimensions. We detail this phenomenon in Appendix F.

These considerations suggest that crosscoders should naturally achieve effective feature alignment, mapping semantically equivalent features from different snapshots to consistent positions in the unified feature space.

## 4 ASSESSING CROSS-SNAPSHOT FEATURES

### 4.1 OVERVIEW OF FEATURE EVOLUTION

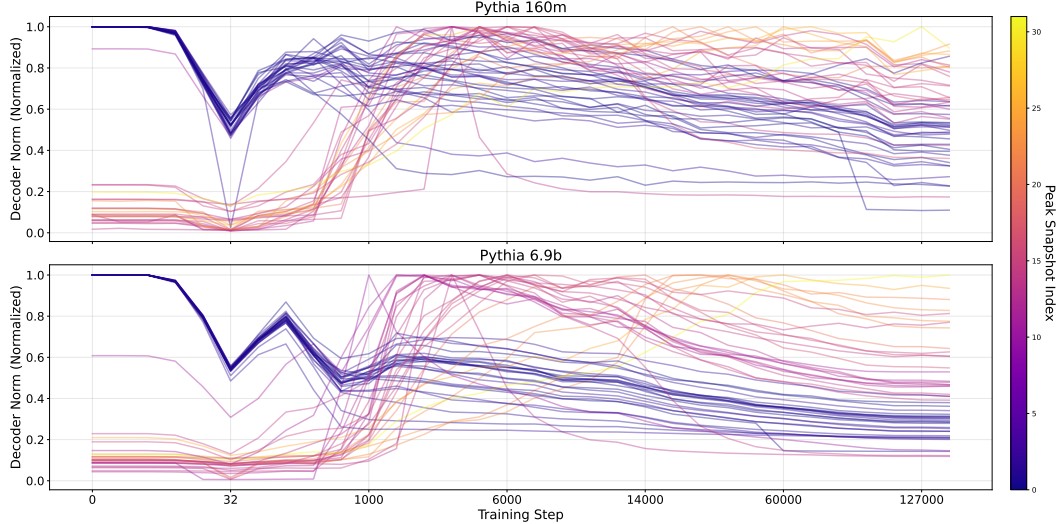

Figure 4: Overview of cross-snapshot feature decoder norm evolution. Features are extracted by a 98,304-feature crosscoder on Pythia-160M (top) and a 32,768-feature crosscoder on Pythia-6.9B (bottom).

Figure 4 shows 50 randomly sampled features and their decoder norm evolution across snapshots. Each feature's decoder norms are linearly rescaled to a maximum of 1. Most cross-snapshot features exhibit two distinct developmental patterns:

1. *Initialization features* that exist from random initialization, exhibit a sudden drop and recovery around step 128, then gradually decay. The existence of these features has been established by Bricken et al. (2023) and Heap et al. (2025).

2. *Emergent features* that begin forming primarily around step 1000, reaching peak intensity at various subsequent training steps. There also exists emergent features only appearing in late training, which we discuss in later sections.

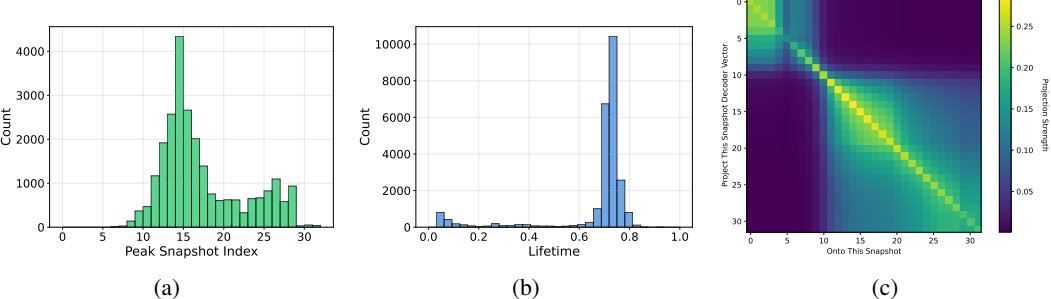

|                  (a)                  |                  (b)                  |                  (c)                  |

Figure 5: Statistics of emergent features in a 98,304-feature crosscoder on Pythia-160M. (a) Distribution of peak emergence times. (b) Distribution of feature lifetime. (c) Mean projection of each feature's decoder vector of snapshot $\theta_i$ onto its decoder vector of snapshot $\theta_j$.

**Feature emergence steepness.** Previous studies suggest that loss curves comprise discrete phase changes, each contributing to different circuits at distinct training stages, with evidence from toy model features (Elhage et al., 2022), 5-digit addition (Nanda et al., 2023), and in-context learning (Olsson et al., 2022). From a feature evolution perspective, we investigate whether features emerge abruptly enough to constitute the fundamental units of these phase transitions. By evaluating the time spent by each feature from initial emergence to peakness, we find that emergence steepness exhibits considerable variation, demonstrating the coexistence of both gradually-formed and abruptly-appearing features.

**Feature persistence after emergence.** We next examine whether emergent features persist after formation. We define the lifetime of a crosscoder feature as $|\{j \mid \|W_{\text{dec},i}^{\theta_j}\| > 0.3\}|$, where a threshold is introduced to block out near-zero decoder norms. Figure 5b shows the lifetimes of all emergent features. We find that most features persist for extended periods (above 60% of snapshots) after formation, indicating that: (1) LLMs retain learned features robustly, and (2) our crosscoders successfully align and track these features across snapshots.

**A universal directional turning point.** To further investigate the geometry of feature evolution, we compute the projections between feature directions across snapshots. The directional evolution patterns prove remarkably consistent: most features undergo drastic directional shifts around step 1,000, rendering pre- and post-step 1,000 directions nearly orthogonal (Figure 5c). Subsequently, features continue to rotate more gradually, with directions at the final snapshot maintaining notable cosine similarities to their initial post-step 1,000 orientations—a significant finding given the high-dimensional activation space.

## 4.2 CORRELATION BETWEEN EMERGENCE STEP AND COMPLEXITY

To gain deeper insights into features emerging at different training stages, we leverage LLMs to automatically assess their complexity based on top activation patterns.

We follow Cunningham & Conerly (2024) to score feature complexity for 100 randomly sampled emergent features based on their top activations, ranging from 1 (simple) to 5 (complex). The

result in Figure 6a reveals a non-trivial correlation with moderate strength between peak timing and complexity scores, suggesting that more complex features tend to emerge later in training. The scoring rubrics and complete prompt for automated complexity scoring can be found in Appendix B.

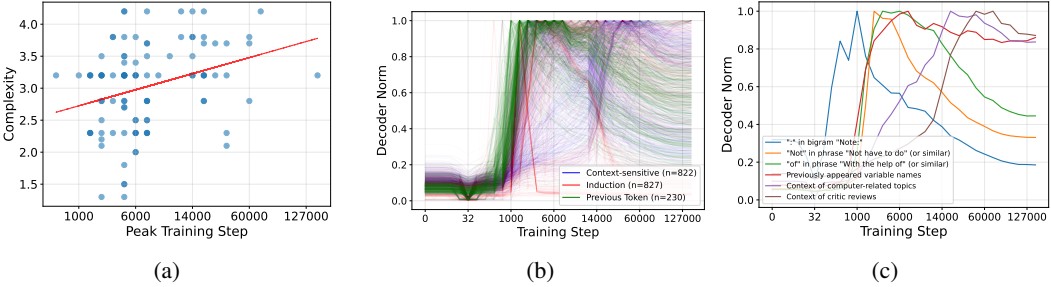

(a)          (b)          (c)

Figure 6: (a) Complexity scores (evaluated by Claude Sonnet 4) versus peak emergence times, showing a moderate positive correlation (Pearson r = 0.309, p = 0.002). (b) Decoder norm evolution trajectories for all previous token, induction, and context-sensitive features across training. (c) More cases of feature decoder norm evolution trajectories.

## 4.3 CASE STUDY ON TYPICAL CROSS-SNAPSHOT FEATURES

We employ simple rule-based approaches to find several well-studied feature types in a crosscoder with 32,768 features trained on Pythia-6.9B, including previous token features (which activate based on the preceding token), induction features (which activate on the second [A] in patterns [A][B]...[A][B] and help predict [B]), and context-sensitive features. These feature categories are extensively documented in prior research on neurons and SAE features (Gurnee et al., 2024; Huben et al., 2024; Ge et al., 2024).

Figure 6b demonstrates the distinct temporal pattern in feature emergence: previous token features arise early (around steps 1,000–5,000), while induction and context-sensitive features appear later and over a wider range (mostly steps 10,000–100,000). This suggests a general emergence hierarchy, from previous token to induction to context-sensitive features, which aligns with their increasing complexity. This finding is also consistent with the causal dependency between induction heads and previous token heads (Olsson et al., 2022).

For the majority of features that cannot fit into specific rule-based patterns, we further demonstrate additional cases of random emergent features with different evolutionary patterns (Figure 6c). Labels are annotated by summarizing the top activation samples.

## 5 CONNECTING MICROSCOPIC EVOLUTION TO MACROSCOPIC BEHAVIORS

One of the ambitious missions of mechanistic interpretability research is to connect feature-level findings with the model's downstream performance. To this end, we employ attribution-based circuit tracing techniques (Syed et al., 2023; Ge et al., 2024; Marks et al., 2025) to investigate the causal effects of crosscoder features formation on downstream task metrics improvements.

**Method.** Let metric $m : \mathbb{R}^{d_{\text{model}}} \to \mathbb{R}$ be an arbitrary scalar-valued function of model activations $a^\theta(x)$.[1] To quantify the causal effect of each feature activation $f_i(x)$ on metric $m$, we first decompose model activations into per-feature representations:

$$a^\theta(x) = \hat{a}^\theta(x) + \epsilon^\theta(x) = \sum_{i=1}^{n_{\text{features}}} f_i(x) \cdot W^\theta_{\text{dec},i} + b^\theta_{\text{dec}} + \epsilon^\theta(x) \tag{3}$$

---

[1]The original notation in Marks et al. (2025) uses $m$ as a function of input $x$, where the computation graph flows through nodes of interest. Since we focus on feature effects within a single layer (e.g., layer 6 of Pythia-160M), we simplify $m$ as a function of model activations to avoid complex intervention notation.

where $\epsilon^\theta(x) \in \mathbb{R}^{d_{\text{model}}}$ represents the crosscoder reconstruction error. This decomposition incorporates crosscoder features into the computation graph, enabling gradient computation with respect to features. We then estimate each feature's causal effect using its attribution score:

$$\text{attr}_i^\theta(x) = f_i(x) \cdot \frac{\partial m(a^\theta(x))}{\partial f_i(x)} \tag{4}$$

where the gradient $\frac{\partial m}{\partial f_i}$ flows through the decomposition in Eq. 3. This attribution score employs first-order Taylor expansion as a linear approximation of model computation.

For structured downstream tasks with clean/corrupted input pairs, such as subject-verb agreement (Finlayson et al., 2021), we can employ the full framework of attribution patching (Syed et al., 2023; Marks et al., 2025) by emphasizing differences between clean and corrupted inputs:

$$\text{attr}_i^\theta(x, \tilde{x}) = [f_i(x) - f_i(\tilde{x})] \cdot \frac{\partial m(a^\theta(x))}{\partial f_i(x)} \tag{5}$$

where $\tilde{x}$ is the corrupted version of input $x$. Attribution patching isolates components explaining the transition from corrupted to clean performance, excluding the majority of components that contribute to baseline model function. This focused approach improves the validity of the linear approximation.

In practice, we employ the integrated gradient (IG) version of the attribution scores defined in Eq. 4 and Eq. 5, which compute gradients at evenly-spaced points from $x$ to $\tilde{x}$ ($x$ to zero vector in the pure attribution case). Details of IG computation can be found in Appendix E.

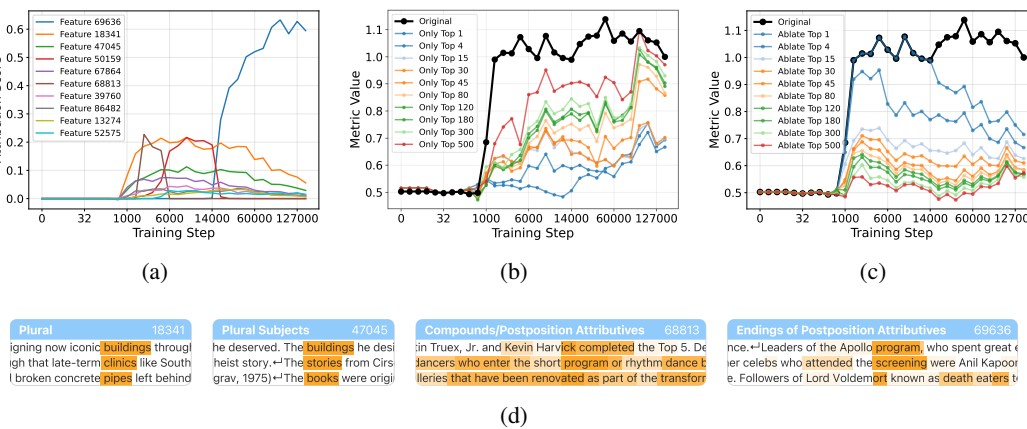

(a)            (b)            (c)

(d)

Figure 7: Crosscoder feature attribution on the Across-PP variant of the subject-verb agreement task, e.g. "The teachers near the desk are". We use a crosscoder train at layer 6 of Pythia-160M with 98,304 features. (a) The attribution scores of top contributing features over time. (b) The metric recovery when ablating all features except the top $k$ contributing features. (c) The metric recovery when ablating the top $k$ contributing features. (d) Top activation samples of key features contributing to this task. Features recognizing plural nouns appear before features recognizing postposition attributives.

**Experimental setup.** We evaluate our method on subject-verb agreement (SVA) (Finlayson et al., 2021), induction, and indirect object identification (IOI) (Wang et al., 2023) tasks, using 1000 samples each to identify critical features.

For SVA and IOI, we create corrupted controls by swapping singular/plural forms (SVA) or altering the second subject (IOI), using logit differences between clean and corrupted answers as metrics. For induction, which lacks natural corruptions, we use target answer log probabilities. We compute IG attribution scores for all features on a 98,304-feature crosscoder trained on Pythia-160M, and rank crosscoder feature contribution by mean attribution scores across all snapshots. We then perform

complementary ablations: (1) removing top-ranked features, and (2) removing all except top-ranked features.

**Results.** Figure 7 shows results for the Across-PP variant of SVA, where postpositional attributives separate subjects and verbs, e.g. "The teachers near the desk are". We identify key contributing features ordered by emergence time: (1) Features 18341 and 47045 capture plural nouns, with 47045 specialized for plural subjects; (2) Feature 68813 marks compound subjects and postpositional attributives; (3) Features 50159 and 69636 identify endings of postpositional attributives, with 69636 showing higher accuracy. Additional features include subject-specific and context-specialized plural markers with lower task contributions. Notably, Features 68813, 50159, and 69636 alternately dominate the metric, revealing circuit-level model evolution through component iteration.

Ablation experiments across all tasks demonstrate that within tens of features, we can consistently disrupt or recover model performance on downstream tasks across training snapshots, confirming our method identifies necessary and sufficient task components.

## 6 OBSERVATIONS OF A STATISTICAL-TO-SUPERPOSITION TRANSITION

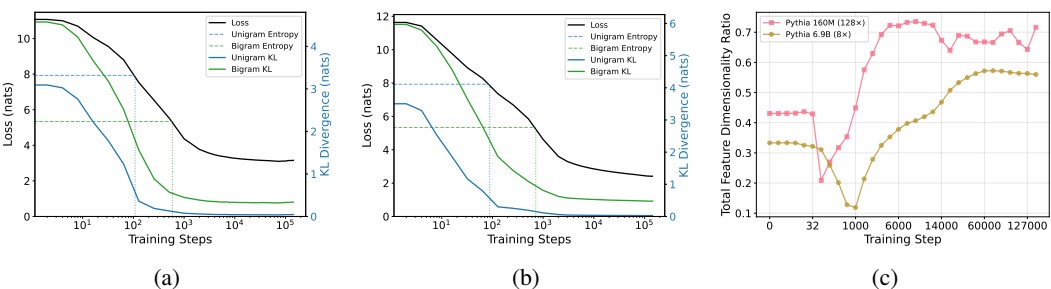

(a)                              (b)                              (c)

Figure 8: Observations supporting a statistical-to-superposition transition in Pythia-160M and Pythia-6.9B. (a) Unigram and bigram KL divergence evolution in Pythia-160M. The convergence timing of KL divergences coincides with when training loss approaches the theoretical minimum (unigram/bigram entropy). (b) Unigram and bigram KL divergence evolution in Pythia-6.9B. (c) Total feature dimensionality ratio (relative to activation space dimension) over training time.

What is the model learning at the beginning of training while the training loss rapidly decreases, if no features are formed at this period? We hypothesize that the rapid loss decrease in early training is driven by learning coarse statistical patterns, rather than by forming distinct features. After this initial optimization nears completion, sparse features emerge in superposition for further loss reduction.

This two-stage structure is deeply consistent with the fitting-to-compressing phase transition predicted by the information bottleneck theory (Shwartz-Ziv & Tishby, 2017).

**Early training almost exclusively learns uni- and bi-gram distributions.** Following previous work on language model learning statistical patterns (Takahashi & Tanaka-Ishii, 2017; Xu et al., 2019; Choshen et al., 2022; Belrose et al., 2024; Svete & Cotterell, 2024; Nguyen, 2024; Chen et al., 2024), we compute the KL divergence between the true token distribution $Q$ and the model's predicted distribution $P$. We randomly sample 10M tokens from SlimPajama to approximate both distributions. We then evaluate unigram KL divergence $D_{KL}(P(x) \parallel Q(x))$ and bigram KL divergence $D_{KL}(P(x_i \mid x_{i-1}) \parallel Q(x_i \mid x_{i-1}))$.

The results show that both unigram and bigram KL divergences rapidly converge to low values during early training (Figure 8a and 8b). Furthermore, the training losses during this period approach the theoretical minimum achievable if the model perfectly matched the true token distributions—i.e., the entropy of these distributions. This suggests that the model primarily learns to fit statistical regularities (Zipf's law (Zipf, 1935; Piantadosi, 2014)) during the early training stage, which explains the dense nature of internal representations at this phase.

**Total feature dimensionality undergoes compression and expansion.** To directly measure superposition status at each snapshot, we adapt the approach from Elhage et al. (2022) and compute the dimensionality of each feature at snapshot $\theta$ as:

$$D_i = \frac{\|W_{\text{dec},i}^\theta\|^2}{\sum_{j=1}^{n_{\text{features}}} \left(\hat{W}_{\text{dec},i}^\theta \cdot W_{\text{dec},j}^\theta\right)^2} \tag{6}$$

where $\hat{W}_{\text{dec},i}^\theta$ is the normalized version of decoder vector $W_{\text{dec},i}^\theta$. In contrast to its original application in toy models with ground-truth features, we apply this metric to crosscoder features due to their consistent alignment across training snapshots, providing insight into superposition dynamics.

We compute total feature dimensionalities for crosscoders trained on Pythia-160M and Pythia-6.9B (Figure 8c). Under ideal superposition, features should form symmetric arrangements with total dimensionalities summing to the activation space dimension. Feature dimensionalities should go down if large interference exists among features. We observe that total feature dimensionality first decreases then increases around the turning point, eventually reaching approximately 70% of available dimensions in the crosscoder with 98,304 features for Pythia-160M. Features from Pythia-6.9B account for a smaller proportion of activation space dimensions, likely due to the limited representational capacity of the 32,768-feature crosscoder. Nevertheless, both models exhibit the same trend, suggesting that initialization features initially form weak superposition, then become compressed to accommodate emergent features. This indicates that the model develops into a feature learning phase.

## 7 LIMITATIONS

**Scope and generalizability.** Superposition has been proven to be a general phenomenon in deep neural networks (Elhage et al., 2022). However, our analysis focuses on feature evolution in Pythia suite models using their open-source training snapshots. While previous research on feature universality (Wang et al., 2025b) suggest our method might be able to generalize to different architectures, datasets, post training dynamics, and tasks beyond language modeling, the extent to which feature evolution patterns are consistent across diverse settings remains to be established. We leave broader generalization studies to future work.

**Limited downstream task complexity.** Section 5 establishes the connection between feature evolution and model behavior. However, the downstream tasks we examine are relatively simple, constrained by multiple factors including Pythia model capabilities, sparse dictionary quality, and the current state of circuit tracing methodologies. Scaling to more complex downstream tasks represents a natural direction for future work.

**Discrete snapshot requirement.** Crosscoder training requires activations from discrete training snapshots, with memory and computational costs scaling linearly with snapshot count, which limits observational granularity. Potential solutions include architectural modifications for online multi-snapshot processing or incorporating gradient information to capture continuous training dynamics.

## 8 CONCLUSION

We introduce crosscoders to study feature evolution in LLM pre-training. Our analysis reveals two patterns: initialization-dependent features and emergent features, with complex patterns emerging later. We establish causal connections between feature evolution and downstream performance. Supported by uni- and bi-gram distribution analysis and feature dimensionality dynamics, we propose that model pre-training can be roughly divided into a statistical learning phase and a feature learning phase. This work bridges mechanistic interpretability with training dynamics.

## REPRODUCIBILITY STATEMENT

All our crosscoders are based on open-source models (Pythia suite) and datasets (SlimPajama) with public accessibility. We provide source code for (1) generating model activations on each snapshot, (2) training crosscoders, and (3) analyzing trained crosscoders at `https://github.com/OpenMOSS/Language-Model-SAEs`. Detailed instructions for replicating our results are provided in the `examples/reproduce_evolution_of_concepts/README.md` file. We note that training large crosscoders requires substantial computational resources and disk space, as also listed in the `README.md` file.

## ACKNOWLEDGEMENT

This work was supported by the National Natural Science Foundation of China (No. 62525602).

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

# A CROSSCODER TRAINING DETAILS

Section 3 presents the mathematical definition of our cross-snapshot crosscoder. However, in practical applications, additional architectural design and training techniques are required to advance the Pareto frontier of sparsity versus reconstruction fidelity. In this section, we detail our selection of activation function $\sigma(\cdot)$ and regularization function $\Omega(\cdot)$, and present our training hyperparameters and their results. We also compare crosscoder performance to standard SAEs to evaluate how well crosscoders perform at sparse dictionary learning.

## A.1 SELECTION OF ACTIVATION FUNCTION AND REGULARIZATION FUNCTION

The sparsity of natural features is the fundamental hypothesis underlying superposition (Olshausen & Field, 1997; Elhage et al., 2022; Huben et al., 2024). To obtain crosscoder features with optimal sparsity for ideal interpretability and monosemanticity while maintaining reconstruction fidelity, we carefully select the activation function $\sigma(\cdot)$ and the regularization function $\Omega(\cdot)$.

Previous SAE studies predominantly use ReLU activation with L1 regularization (Bricken et al., 2023; Huben et al., 2024). However, this configuration produces weak feature activations that are largely noise, compromising both interpretability and sparsity. We address this by adopting JumpReLU (Rajamanoharan et al., 2024) as the activation function, which eliminates activations below learned thresholds (trained via straight-through estimation (Bengio et al., 2013)).

To prevent features from becoming permanently inactive at certain snapshots, we incorporate decoder norms into the activation decision. Given pre-activation $z(x)$:

$$z(x) = \sum_{\theta \in \Theta} W_{\text{enc}}^{\theta} a^{\theta}(x) + b_{\text{enc}} \tag{7}$$

The $i$-th feature activation at snapshot $\theta$ is defined as:

$$f_i^{\theta}(x) = z_i(x) \cdot H(z_i(x) \cdot \|W_{\text{dec},i}^{\theta}\| - t_i) \tag{8}$$

where $H(\cdot)$ is the Heaviside step function and $t_i \in \mathbb{R}$ is the JumpReLU threshold for feature $i$.

This design ensures that features with small decoder norms require stronger pre-activations to activate, preventing complete feature death while maintaining sparsity. Although this means truly inactive features retain small positive decoder norms rather than zero values, this architectural choice significantly improves crosscoder performance and enables better feature tracking across snapshots.

For regularization, we employ a combination of tanh and quadratic frequency penalties (Smith et al., 2025): the tanh component provides a superior L0 approximation by reducing penalties on strong activations, while the quadratic frequency penalty suppresses high-frequency features. This yields the following batched sparsity loss:

$$\omega_i^{\theta}(\mathcal{B}) = \frac{1}{\mathcal{B}} \sum_{x \in \mathcal{B}} \tanh\left(f_i(x) \cdot \|W_{\text{dec},i}^{\theta}\|\right)$$
$$\mathcal{L}_{\text{sparsity}}(\mathcal{B}) = \lambda_{\text{sparsity}} \sum_{\theta \in \Theta} \sum_{i=1}^{n_{\text{features}}} \omega_i^{\theta}(\mathcal{B}) \cdot \left(1 + \frac{\omega_i^{\theta}(\mathcal{B})}{\omega_0}\right) \tag{9}$$

where $\mathcal{B} \subset \mathcal{C} \times \mathbb{N}$ is an input batch, and $\omega_i^{\theta}(\mathcal{B})$ is the single-batch differentiable estimate for the activation frequency on snapshot $\theta$ of $i$-th crosscoder feature, using tanh as L0 approximation. A new hyperparameter $\omega_0$ is introduced to quadratically penalize feature activation when $\omega_i^{\theta}(\mathcal{B}) \gg \omega_0$.

## A.2 SELECTION OF PRE-TRAINING SNAPSHOTS

To balance training cost with the granularity of feature evolution analysis, we train crosscoders using 32 source snapshots from the 154 open-source snapshots in the Pythia suite. Figure 10 shows the training steps of the selected snapshots.

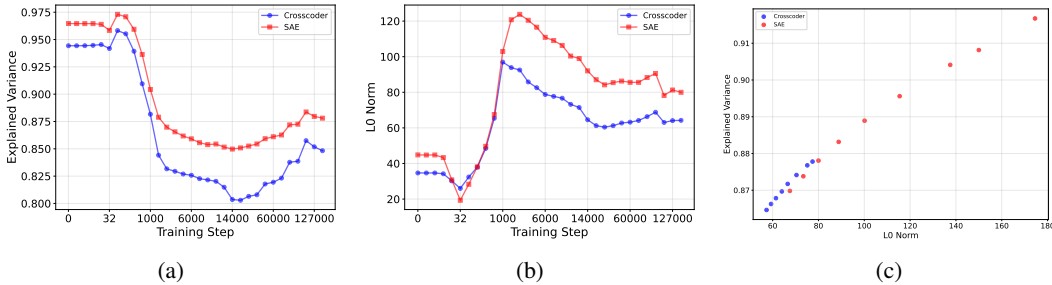

(a)                                (b)                                (c)

Figure 9: Comparison between crosscoders and per-snapshot SAEs. (a) The explained variance of crosscoders versus SAEs at each snapshot. (b) The L0 norm of crosscoders versus SAEs at each snapshot. (c) The Pareto frontier comparison of crosscoders and SAEs trained on the final snapshot.

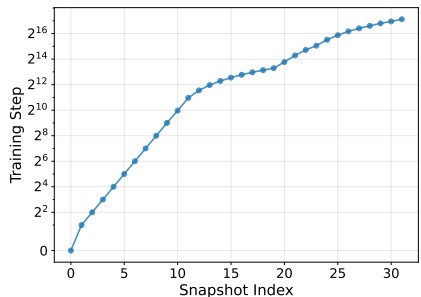

Figure 10: Selected snapshots from Pythia suite.

### A.3 DISTRIBUTED TRAINING STRATEGY FOR CROSSCODERS

Crosscoders require parameters that scale with the number of source snapshots $n_{\text{snapshots}}$, resulting in significantly higher memory and computational requirements compared to SAEs with equivalent feature counts, particularly when using many source snapshots.

To efficiently train crosscoders, we employ a **head parallelism** distributed training strategy, a variant of tensor parallelism (Shoeybi et al., 2019). With $k$ processes where $k$ divides $n_{\text{snapshots}}$, each process handles encoding and decoding for $\frac{n_{\text{snapshots}}}{k}$ source snapshots. Pre-activations are computed via All-Reduce operations (Sanders et al., 2019). Unlike standard tensor parallelism, our approach processes activations from each snapshot separately on individual processes, reducing I/O overhead for reading activations from disk.

### A.4 EXPERIMENTS

We train crosscoders on Pythia-160M and Pythia-6.9B snapshots at various scales (6,144 to 98,304 features on Pythia-160M, and 32,768 features on Pythia-6.9B). We primarily focus on middle layers (Layer 6 in Pythia-160M, and Layer 16 in Pythia-6.9B)[2], but also train crosscoders of 24,576 features on all layers of Pythia-160M for comprehensive analysis.

All crosscoders are trained on 800M tokens from the SlimPajama corpus using the Adam optimizer (Kingma & Ba, 2017) with $\beta$ values of $(0.9, 0.999)$. To prevent straight-through estimation from causing rapid threshold increases, we apply a reduced learning rate to JumpReLU threshold updates via a multiplier on the global learning rate. JumpReLU thresholds are initialized to $0.1$ for all features. The learning rate schedule includes 10% warm-up steps followed by 20% decay steps. We initialize encoders as transposes of their corresponding decoders, with identical initialization matrices across all snapshots. We employ initialization search to identify optimal decoder norms that minimize loss at initialization. Additional hyperparameters are listed in Table 1.

---

[2]By "Layer $i$", we refer to the activations at the output of the $i$-th transformer layer.

Table 1: Hyperparameters for crosscoder training

| Parameter | Pythia-160M | Pythia-6.9B |
|---|---|---|
| Learning Rate | 5e-5 | 1e-5 |
| Batch Size | 2048 | 2048 |
| Feature Expansion Ratio | 8×, 16×, 32×, 64×, 128× | 8× |
| Sparsity Coefficient ($\lambda_{\text{sparsity}}$) | 0.3 | 0.3 |
| JumpReLU Threshold LR Multiplier | 0.1 | 0.3 |

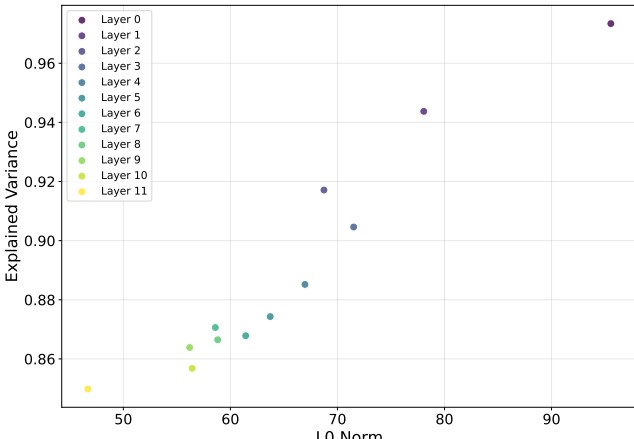

Figure 11: Explained variance versus L0 norm of crosscoders trained on all layers (0-11) of Pythia 160M

## A.5 RESULTS ON OTHER LAYERS

We train additional crosscoders with 24,576 features on all layers (0-11) of Pythia-160M using the same hyperparameters as Layer 6 (Table 1). Figure 11 shows the explained variance and L0 norm for these crosscoders, which exhibit similar performance but different trade-offs between sparsity and reconstruction quality.

To demonstrate feature evolution across all layers, we apply the same visualization strategy from Section 4, plotting decoder norms for 50 randomly selected features per layer (Figures 12 and 13). Most middle layers exhibit the same evolutionary patterns as Layer 6, supporting our findings in Section 4. We would like to note that the majority of features in Layers 0 and 1 exist from initialization, which aligns with the observation that early layers implements more low-level, especially single-token features (He et al., 2024).

## B AUTOMATED SCORING OF COMPLEXITY

We leverage Claude Sonnet 4 for automated complexity score evaluation. For each feature, we select the top 10 activating samples with 100 surrounding tokens around the strongest activating tokens and apply the following prompt:

```
# Neural Network Feature Analysis Instructions
We're analyzing features in a neural network. Each feature activates on
specific words, substrings, or concepts within short documents. Activating
words are marked as `<<text, {activation}>>` where `{activation}` indicates
the strength of activation (higher values = stronger activation). You'll
receive documents containing highest activations tokens and tokens
surrounding them.

## Your task:
### 1. Summarize the Activation (<20 words)
```

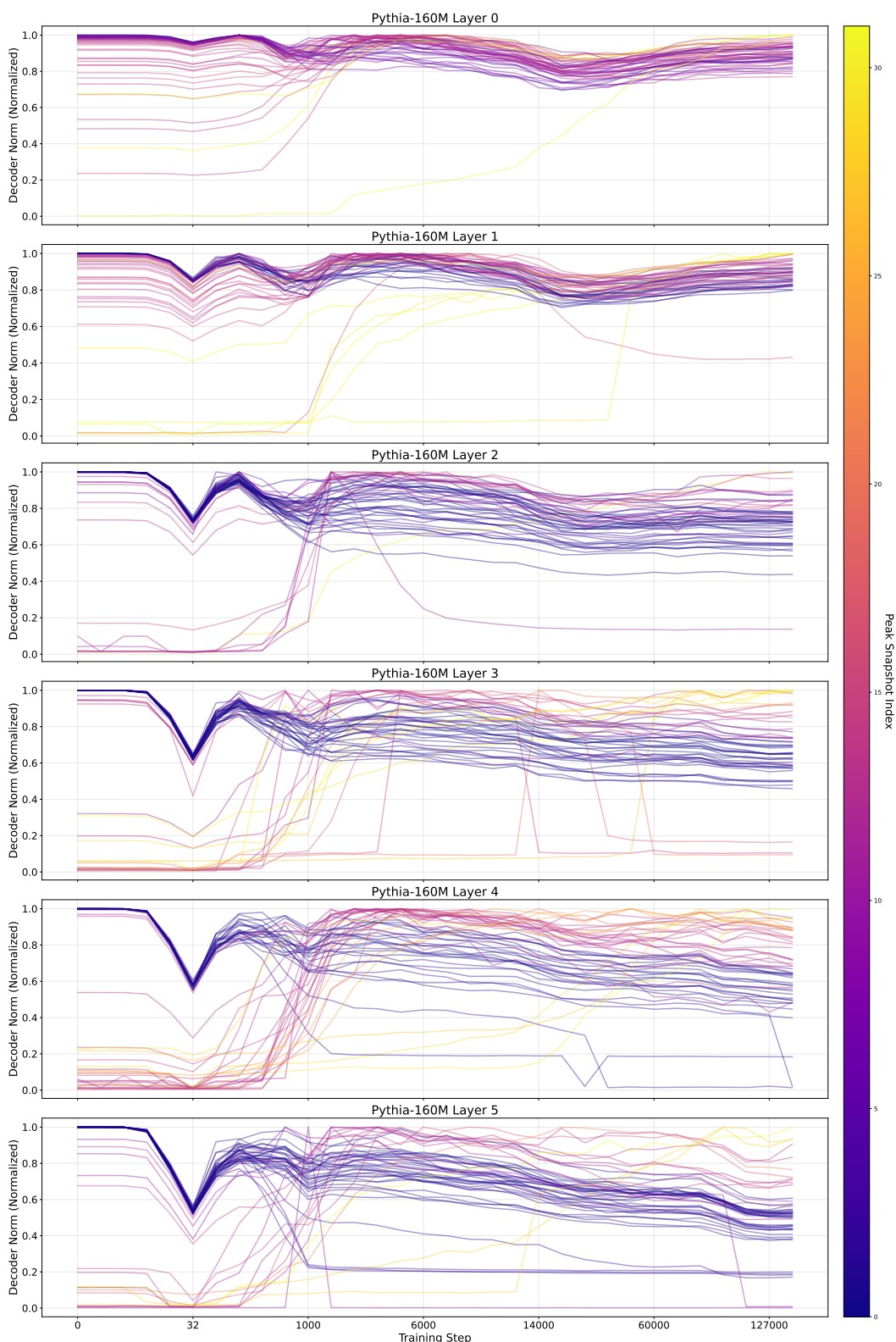

Figure 12: Feature decoder norm evolution of Layer 0 to Layer 5 in a 24,576-feature crosscoder trained on Pythia-160M.

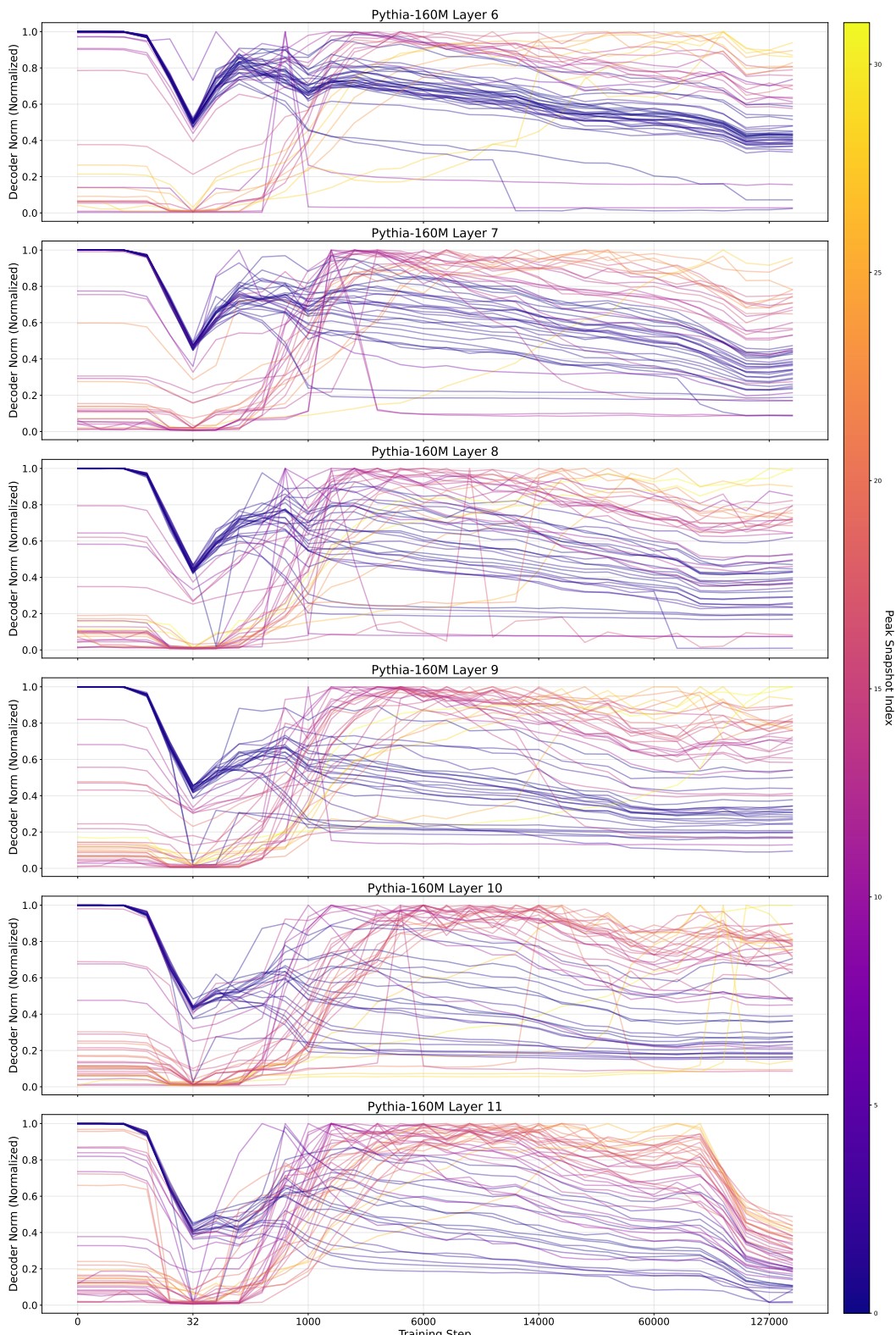

Figure 13: Feature decoder norm evolution of Layer 6 to Layer 11 in a 24,576-feature crosscoder trained on Pythia-160M.

```
Examine the marked activations and summarize what the feature detects in
one sentence.
- Avoid being overly specific|your explanation should cover most/all
  activating words
- If all words in a sentence activate, focus on the sentence's concept
  rather than individual words
- Note relevant patterns in capitalization, punctuation, or formatting
- Prioritize strongly activated tokens
- Keep explanations simple and concise
- Avoid long word lists

### 2. Assess Feature Complexity (1-5 scale, with decimal precision allowed)
Rate the feature's complexity:
- **5**: Rich feature with diverse contexts unified by an interesting theme
- **4**: High-level semantic structure with potentially dense activation
- **3**: Moderate complexity|phrases, categories, or sentence structures
- **2**: Synonyms or words at a same class
- **1**: Single specific word or token
You may use decimal values (e.g., 3.7) for more precise assessment.

### Output Format
Your output should be in JSON format, with two fields: summarization and
complexity. You should directly output the JSON object, without any other
text.
```

## C  ASSESSING CROSSCODER DECODER NORM

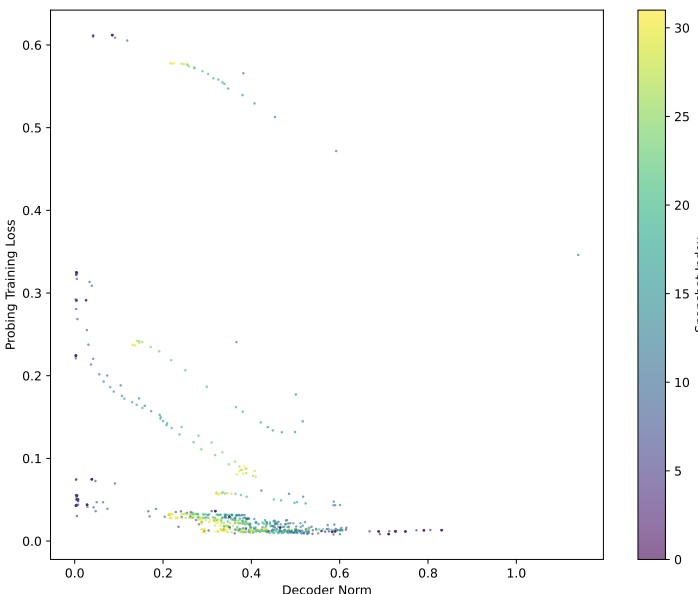

Figure 14: Linear probe training errors versus feature decoder norms of 20 randomly sampled features from a crosscoder with 6,144 features on Pythia-160M.

Section 4 takes advantage of the crosscoder decoder norms to study feature evolution. However, are the crosscoder decoder norms well quantitative indicators of the extent to which features evolved? Aside from theoretical statements, we conduct experiments to examine whether crosscoder decoder norms can reflect the intensity of features. We leverage linear probing to test the correlation between the feature decoder norm and the linear separability of the feature activations on each crosscoder feature.

We train linear probing classifiers (referred to as probes) separately on activations of each model snapshot to classify whether each crosscoder feature activates. For each feature $i$ and each snapshot $\theta$, our probes map model activations to the predicted feature activating probability $p_i^\theta(x)$ as:

$$p_i^\theta(x) = \text{sigmoid}(w_{\text{probe},i}^\theta \cdot a^\theta(x) + b_{\text{probe},i}^\theta) \tag{10}$$

where $w_{\text{probe},i}^\theta \in \mathbb{R}^{d_{\text{model}}}$ and $b_{\text{probe},i}^\theta \in \mathbb{R}$ is the weight and bias of the probe w.r.t. feature $i$ and snapshot $\theta$. Each probe is trained to minimize the binary cross-entropy loss $y_i \cdot \log p_i^\theta(x) + (1 - y_i) \cdot \big( \log(1 - p_i^\theta(x)) \big)$, given the label $y_i = \text{sgn}\,(f_i(x))$. The training loss of each probe should be a direct measure of the linear separability of the feature activations.

We train probes for 100M tokens for each feature in a 6,144-feature crosscoder on Pythia-160M. The probe errors of each feature show a mean Pearson correlation with the corresponding decoder norms by $-0.867$, with a standard deviation of $0.153$. We demonstrate example probe errors versus feature decoder norms of 20 randomly sampled features in Figure 14. This indicates a strong negative linear relationship between probe errors and crosscoder decoder norms, demonstrating the effectiveness of the crosscoder decoder norms as indicators of feature evolution.

## D    RULES FOR FINDING TYPICAL CROSS-SNAPSHOT FEATURES

We define the rules used to identify previous token features, induction features, and context-sensitive features as follows:

1. **Previous Token Features:** We collect the directly preceding tokens of all activating tokens in the top 20 activating samples and assess their consistency. Token consistency is defined as the proportion of the largest single group when all tokens are normalized by stemming (removing leading and trailing spaces and ignoring case). To exclude bigram or multigram features, we also evaluate the consistency of the activating tokens themselves. A feature is classified as a previous token feature if it exhibits high consistency in previous tokens (above 0.8) and low consistency in activating tokens (below 0.3).

2. **Induction Features:** Induction features should activate on the second [A] in patterns [A][B]...[A][B]. For each activating token [A], we collect its following token [B] and search for previous occurrences of the bigram [A][B]. To distinguish induction features from simpler features that merely activate on any bigram [A][B], we require that the feature does not activate on the first appearance of [A][B]. A feature exhibiting this behavior in at least 20 instances within the top 20 activating samples is classified as an induction feature.

3. **Context-sensitive Features:** We identify context-sensitive features using a simpler rule based on activation density within specific contexts. Context-sensitive features should activate frequently in highly specific contexts, so we require features to have high activation counts within the top 20 activating samples (exceeding 4,000 activations). To exclude features that activate ubiquitously (such as positional or bias features), we filter out features with excessive total activations (below 2M total activations across 100M analyzed tokens).

## E    DETAILS IN DOWNSTREAM TASK ATTRIBUTION

### E.1    FORMALIZATION OF IG ATTRIBUTION SCORE

To more accurately estimate the causal effect of features, we employ the integrated gradient (IG) version of the attribution score (Sundararajan et al., 2017; Hanna et al., 2024; Marks et al., 2025). IG attribution computes gradients along an interpolation path between baseline and target activations, providing more robust estimates than single-point gradients.

For the standard attribution score without clean/corrupted input pairs, we compute the IG version as:

$$\text{attr}_{\text{ig},i}^\theta(x) = \frac{1}{N} \sum_\alpha f_i(x) \cdot \frac{\partial m(a^\theta(x))}{\partial(\alpha f_i(x))} \tag{11}$$

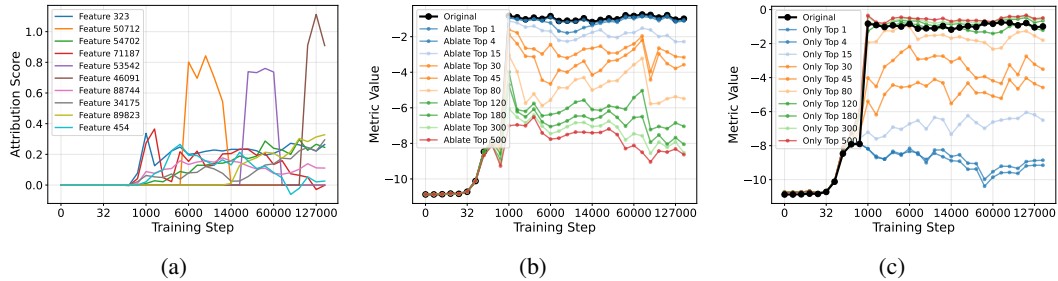

Figure 15: Crosscoder feature attribution on the induction task, using a crosscoder with 98,304 features trained on Pythia-160M.

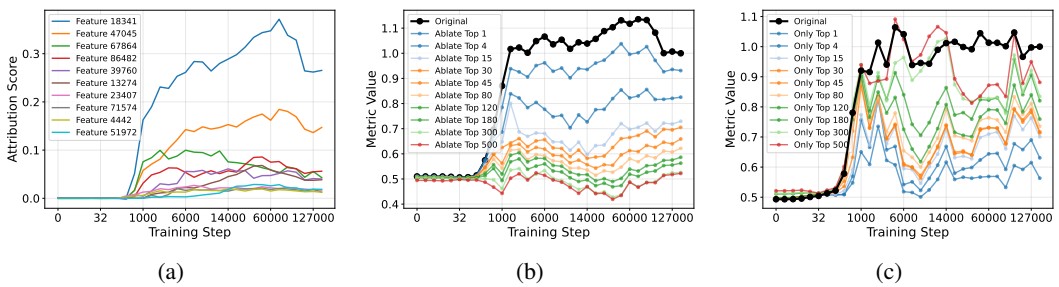

Figure 16: Crosscoder feature attribution on the Simple variant of the SVA task, using a crosscoder with 98,304 features trained on Pythia-160M.

For attribution patching with clean/corrupted input pairs, the IG version is:

$$\text{attr}_{\text{ig},i}^{\theta}(x, \tilde{x}) = \frac{1}{N} \sum_{\alpha} \left[ f_i(x) - f_i(\tilde{x}) \right] \cdot \frac{\partial m(a^{\theta}(x))}{\partial(\alpha f_i(x) + (1 - \alpha) f_i(\tilde{x}))} \qquad (12)$$

where $\alpha \in \{0, \frac{1}{N}, \dots, \frac{N-1}{N}\}$ linearly interpolates between the baseline and target feature activations. Consistent with Marks et al. (2025), we use $N = 10$ interpolation steps for the IG attribution score.

### E.2    INDUCTION TASK

Transformers exhibit in-context learning capabilities (Brown et al., 2020; Zhao et al., 2021; Gao et al., 2021) through induction heads (Olsson et al., 2022)—circuits that look back over the sequence for previous instances of the current token (A), identify the subsequent token (B), and predict the same completion will occur again (forming sequences [A][B] ... [A] → [B]).

To evaluate models' induction abilities and trace them at the feature level, we construct samples with random tokens where identical patterns appear in the middle and at the end of sequences. We test whether models can correctly copy previous text as the next token. For precise feature-level analysis, we restrict next tokens to single capital letters with leading spaces. Since the induction task lacks natural corrupted counterparts, we use the log probability of the correct token as our evaluation metric. Results are shown in Figure 15.

### E.3    SUBJECT-VERB AGREEMENT TASKS

Subject-verb agreement tasks evaluate whether models can predict verbs in the appropriate grammatical form based on their subjects. We test four variants: (1) Simple—the verb directly follows the subject, e.g., "The parents are"; (2) Across-RC—a relative clause intervenes between subject and verb, e.g., "The athlete that the managers like does"; (3) Within-RC—both subject and verb appear within the relative clause, e.g., "The athlete that the managers like"; (4) Across-PP—a prepositional

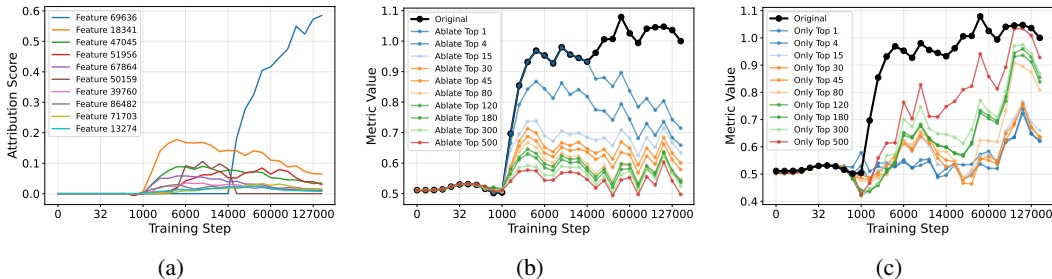

Figure 17: Crosscoder feature attribution on the Across-RC variant of the SVA task, using a cross-coder with 98,304 features trained on Pythia-160M.

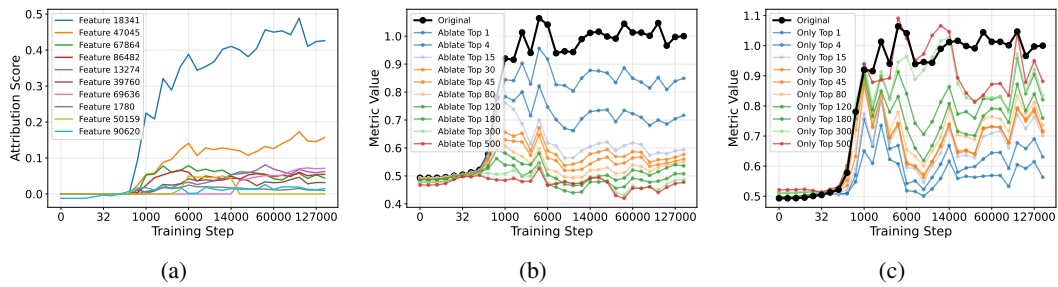

Figure 18: Crosscoder feature attribution on the Within-RC variant of the SVA task, using a cross-coder with 98,304 features trained on Pythia-160M.

phrase separates the subject and verb, e.g., " The secretaries near the cars have". We use data provided by Marks et al. (2025), sampling 1000 examples from each variant. We use the verb in the wrong form as the counterpart in attribution patching. Results are shown in Figure 7, 16, 17, and 18.

### E.4  INDIRECT OBJECT IDENTIFICATION TASK

Indirect Object Identification (IOI) evaluates whether models can correctly predict the indirect object (IO) based on repeated occurrence of subjects (Wang et al., 2023). In IOI tasks, sentences such as "When Mary and John went to the store, John gave a drink to" should be completed with "Mary." We generate IOI samples following the same strategy as Wang et al. (2023), using template sentences with random person names. We use the subject name as the corrupted counterpart in attribution patching. Results are shown in Figure 19.

Note that while ablating top features significantly degrades performance, we cannot recover the original metric using only these top features. This likely occurs because IOI is a complex task requiring multiple feature interactions. The features that distinguish between indirect objects and subjects represent only part of the full computational requirements, making isolated feature sets insufficient for complete task execution.

### E.5  RANDOM BASELINE OF ABLATION

In Figure 20, we add baselines to Figure 7 by ablating/preserving $k$ random features. The results show that our feature selected by attribution patching is far more effective.

## F  CROSSCODER FEATURE SPLITTING

Feature splitting is a well-known phenomenon in SAEs where increasing the dictionary size $n_{\text{features}}$ causes features from smaller SAEs to fragment into multiple distinct features. Under feature splitting, a single concept may not be represented by one feature, but rather by multiple specialized features that activate on the same concept in different contexts.

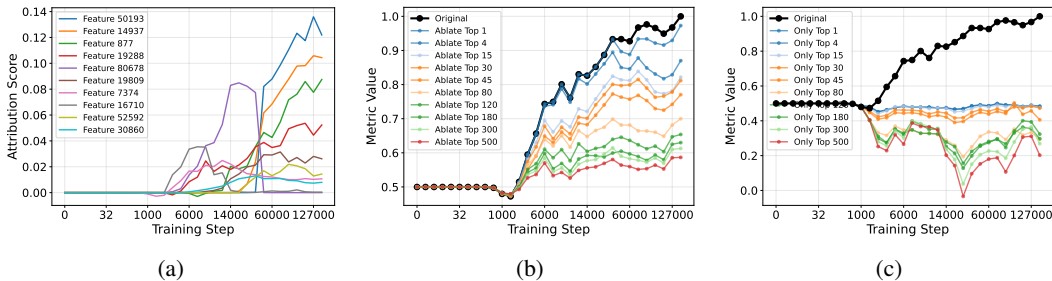

(a)  (b)  (c)

Figure 19: Crosscoder feature attribution on the IOI task, using a crosscoder with 98,304 features trained on Pythia-160M.

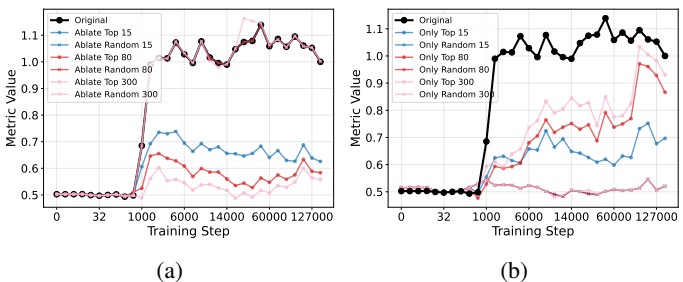

(a)  (b)

Figure 20: Crosscoder feature attribution on the Across-PP variant of the SVA task, with random baselines.

We observe similar feature splitting phenomenon in crosscoders, but across the temporal dimension. We find in rare cases, features with similar semanticity and direction in different snapshots may not be encoded in the same latent of crosscoder feature space, but result in separate features.

For example, we identify feature 66688, 53542, and 42307 that all activate on long sequences containing repeated text patterns, contributing to the induction task (Figures 21, 22, and 23). Each feature exists only during non-overlapping continuous periods across snapshots, with a drastic emergence and disappearance.

To examine whether feature splitting across snapshots relates to dictionary size, we search for feature decoder vectors (across all snapshots) with cosine similarity above 0.7—a high threshold in such high-dimensional space—in crosscoders with varying $n_{\text{features}}$. Across dictionary sizes ranging from 6,144 to 98,304 features, we observe that while each snapshot after step 2,000 activates almost exactly one feature representing this concept, the total number of distinct crosscoder features increases with dictionary size (Figure 24). This confirms that larger dictionaries lead to temporal feature splitting, where a single underlying concept splits into multiple features active at different training stages.

We also observe that temporal feature splitting predominantly occurs among densely activating features, i.e., features that activate frequently across many contexts. This crosscoder feature splitting likely arises from subtly different activation patterns that emerge over training. These findings suggest that language model features may evolve by refining their activation patterns, leading to more specialized representations that warrant separate feature assignments at different training stages.

## G  GENERALIZABILITY STUDIES

In this section, we include more experiments on training and analyzing crosscoders on different base models/initialization/datasets and different numbers of snapshot selection, to examine the generalizablity of our methods.

method is not threadsafe. An endpoint should be configured when the client is created and before any↵   * service requests are made. Changing it afterwards creates inevitable race conditions for any service requests in↵   * transit or retrying.↵   *↵   * @param endpoint↵   *      The endpoint (ex: "cognito-idp.us-east-1.amazonaws.com") or a full URL, including the protocol (ex:↵   *      "https://cognito-idp.us-east-1.amazonaws.com") of the region specific AWS endpoint this client will↵   *      communicate with.↵   * @deprecated use {@link AwsClientBuilder#setEndpointConfiguration(AwsClientBuilder.EndpointConfiguration)} for↵   *      example:↵   *      {@code builder.setEndpointConfiguration(new EndpointConfiguration(endpoint, signingRegion));}↵   */↵   @Deprecated↵   void setEndpoint(String endpoint);↵   /**↵   * An alternative to {@link AWSCognitoIdentityProvider#setEndpoint(String)}, sets the regional endpoint for this↵   * client's service calls. Callers can use this method to control which AWS region they want to work with.↵   * <p>↵   * By default, all service endpoints in all regions use the https protocol. To use http instead, specify it in the↵   * {@link ClientConfiguration} supplied at construction.↵   * <p>↵   * This method is not threadsafe. A region should be configured when the client is created and before any service↵   * requests are made. Changing it afterwards creates inevitable race conditions for any service requests in transit↵   * or retrying.↵   *↵   * @param region↵   *      The region this client will communicate with. See {@link Region#getRegion(com.amazonaws.regions.Regions)}↵   *

Figure 21: Top activation of Feature 66688 in the 98,304-feature crosscoder on Pythia-160M.

Angry Birds Space Mod APK 2022 v2.2.15 (latest Version, Unlimited Money)↵This game is the ultimate in family fun. It's easy to learn, it's challenging enough for adults and kids alike, and it has hours of replay value. The best part? You can play Angry Birds Space Mod APK on your phone or tablet with no need for a computer download! So if you love this game as much as we do and who doesn't? -don't wait another moment before downloading Angry Birds Space Mod APK today.↵Rovio Entertainment CorporationUpdated↵2 days ago Size↵Android 4.1Get it on↵1 Angry Birds Space Mod APK Unlimited Everything↵2 Amazing Features of Angry Birds Space Mod APK↵2.1 Red Bird↵2.2 Yellow Bird↵2.3 Blue Bird↵2.4 White Bird↵2.5 Black Bird↵2.6 Boomerang Bird↵2.7 Pig Soldier↵2.8 Bird Sling↵2.9 Slow Motion↵2.10 Hitting Obstacles with A Slingshot↵2.11 Obstacles in Space↵3 FAQs of Angry Birds Space Mod APK↵3.1 Does Angry Birds Space Mod APK Works Offline?↵3.2 Is It Ads-Free?↵3.3 Is It Easy to Play?↵3.4 Can I play offline without an internet connection on my laptop, PC, Mac, or smartphone?↵3.5 Do I need to root my Android device?↵Angry Birds Space Mod APK Unlimited Everything↵Angry Birds Space Mod APK is the ultimate in family fun. It's easy to learn, it's challenging enough for adults and kids alike, and it has hours of replay value. The best part? You can play Angry Birds Space APK on your phone or tablet with no need for a computer download. So if you love this game as much as we do-and who doesn't? don't wait another moment before downloading Angry Birds Space Mod APK today.↵Download Angry Birds Rio Mod APK↵Download Adventure

Figure 22: Top activation of Feature 53542 in the 98,304-feature crosscoder on Pythia-160M.

## G.1 GENERALIZATION ACROSS MODEL FAMILIES/SEEDS

We conduct experiments on two additional Pythia 160M models with different random seeds (Pythia 160M Seed1 and Seed2) (van der Wal et al., 2025) and the Alias run of Stanford CRFM's GPT-2 model [3]. All of these models have an activation space of 768 dimensions. We train crosscoders of 24,576 features (32x) on Layer 6 of them, and inspect their feature evolution.

**Different initializations (Pythia 160M variants).** We observe highly consistent results across differently initialized versions of Pythia 160M. Decoder norms of initialization features follow identical trajectories, and emergent features begin rising around step 1000, exhibiting a clear two-phase pattern with the same transition point (Figure 26 and 27). Attribution experiments confirm that we successfully identify sparse crosscoder features with significant contributions to downstream tasks (Figure 29 and 30).

**Different base model (GPT-2).** For Stanford CRFM's GPT-2 model, we observe a clear two-phase pattern, with emergent features rising around step 1,000. Upon manual inspection of these features, we still find that initialization features exhibit token-level information, while more complex patterns emerge in emergent features.

However, we find a notable difference: due to absolute positional embeddings, initialization feature norms do not peak at the beginning but rather at later steps (with above-threshold norm at beginning). This difference arises because the absolute positional embeddings initially exhibit large norms (larger than the word embeddings) and dominate the activation norm. Since we normalize all activation sources to the same average activation norm $\sqrt{d_{\text{model}}}$, this large positional contribution substantially reduces the observed norms of initialization features. (Figure 28).

To sum up, the statistical-to-feature-learning phase transition still exists across models, despite small differences. Attribution experiments (Figure 31) again confirm the effectiveness of sparse crosscoder features, with generally similar feature evolution patterns for specific downstream tasks.

**Other models with accessible checkpoints.** We also investigated other model families, including BLOOM (Scao et al., 2022) and OLMo (OLMo et al., 2024). Unfortunately, these models either lack sufficient intermediate checkpoints for fine-grained analysis of pretraining dynamics (BLOOM), or lack sufficiently early checkpoints to capture the two-stage transition (BLOOM and OLMo).

- <a data-lity="message-video" href="https://www.youtube.com/watch?v=Q-C4qqsgs8w" rel="noopener noreferrer nofollow">Power Rangers (2017 Movie) Official Teaser Trailer – 'Discover The Power'</a> [YouTube] 
 
SABAN'S POWER RANGERS follows five ordinary teens who must become something extraordinary when they learn that their small town of Angel Grove — and the world — is on the verge of being obliterated by an alien threat. Chosen by destiny, our heroes quickly discover they are the only ones who can save the planet. But to do so, they will have to overcome their real-life issues and before it's too late, band together as the Power Rangers. 
 
Pretty typical post-Nolan franchise reboot on Lionsgate money. Doesn't really tap into Power Rangers in any enticing way. Not very impressive. Future trailers might excite more.↵– Power Rangers (2017 Movie) Official Teaser Trailer – 'Discover The Power' [YouTube]↵SABAN'S POWER RANGERS follows five ordinary teens who must become something extraordinary when they learn that their small town of Angel Grove — and the world — is on the verge of being obliterated by an alien threat. Chosen by destiny, our heroes quickly discover they are the only ones who can save the planet. But to do so, they will have to overcome their real-life issues and before it's too late, band together as the Power Rangers.↵Pretty typical post-Nolan franchise reboot on Lionsgate money. Doesn't really tap into Power Rangers in any enticing way. Not very impressive. Future trailers might excite more.↵Trailer looks awesome, I like the humor, I like the tone. It's breakfast club meets Chronicle, with an Iron

Figure 23: Top activation of Feature 42307 in the 98,304-feature crosscoder on Pythia-160M.

## G.2 GENERALIZABILITY ACROSS DATASETS

We conduct experiments on the original Pythia 160M snapshots, but train and analyze 24,576-feature crosscoders on the FineWeb-Edu dataset (Penedo et al., 2024), which consists of educational webpages. Our results (Figure 32) show nearly identical patterns to those observed with Pythia 160M crosscoders trained on SlimPajama, exhibiting very similar feature evolution dynamics.

## G.3 SENSITIVITY TO SNAPSHOT SELECTION

All of our above experiments train crosscoders on 32 pre-training snapshots of the language model. To further examine whether snapshot selection affects the trained crosscoder and the captured feature evolution, we train a 24,576-feature crosscoder on 16 pre-training snapshots of Pythia 160M, downsampled from the original set by selecting every other snapshot. The result (Figure 33) shows that the overall feature evolution exhibits nearly identical trends, with only a reduction in temporal resolution.

## H MORE EXAMPLES OF FEATURE

We additionally list some emergent features and demonstrate their top activating samples and decoder norm evolution in Figure 34.

## I THE USE OF LARGE LANGUAGE MODELS

The use of Large Language Models as an assistive tool in this paper is limited to the following two aspects:

1. The automated generation of feature complexity scores, as detailed in Section 4.2 and Appendix B.
2. Grammar correction and stylistic refinement of the manuscript.

---

[3]Obtained from `https://huggingface.co/stanford-crfm/alias-gpt2-small-x21`.

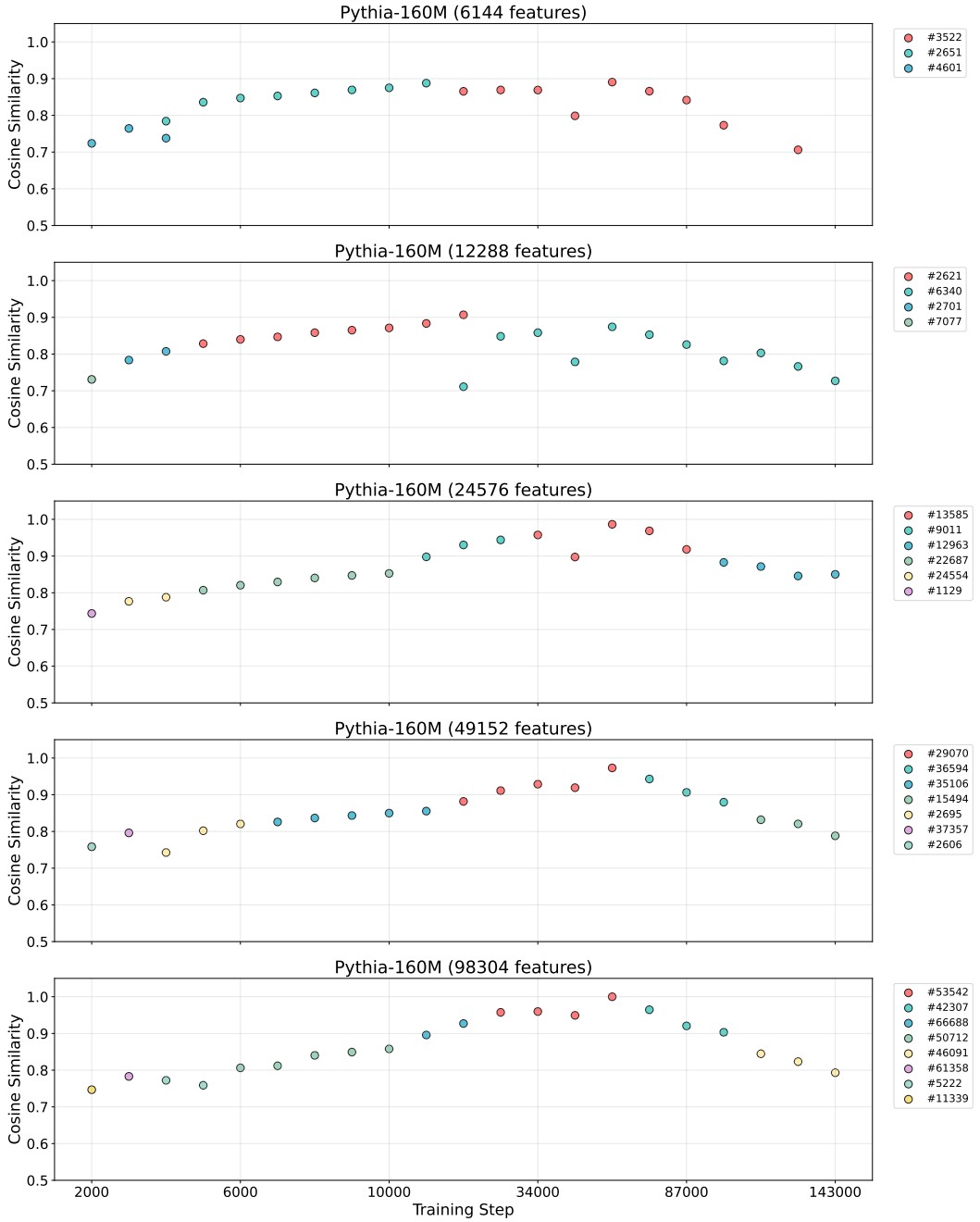

Figure 24: Crosscoder features similar to Feature 53542 of a 98,304-feature crosscoder. The number of features split increases with dictionary size.

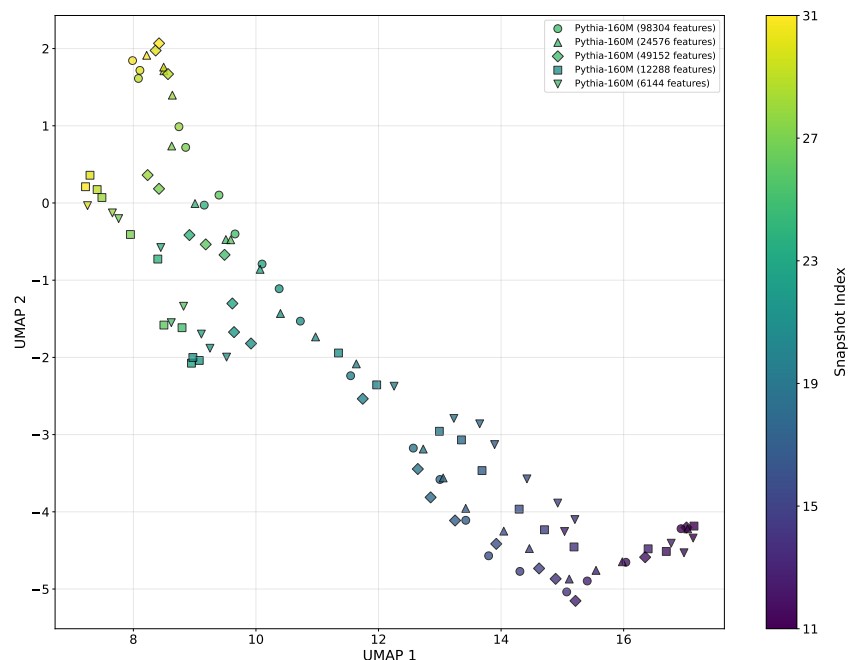

Figure 25: UMAP visualization of features shown in Figure 24.

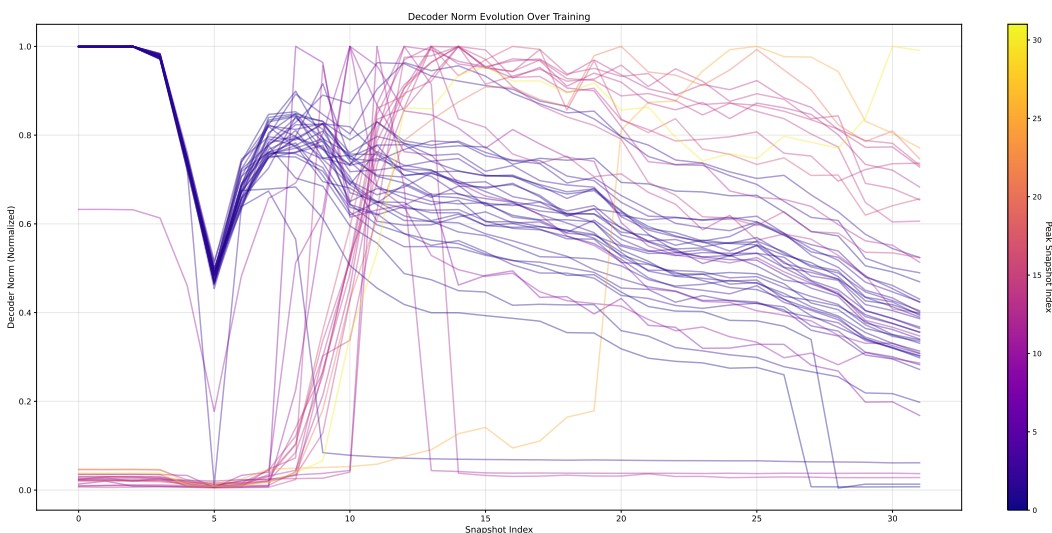

Figure 26: Overview of cross-snapshot feature decoder norm evolution in Pythia-160M Seed1.

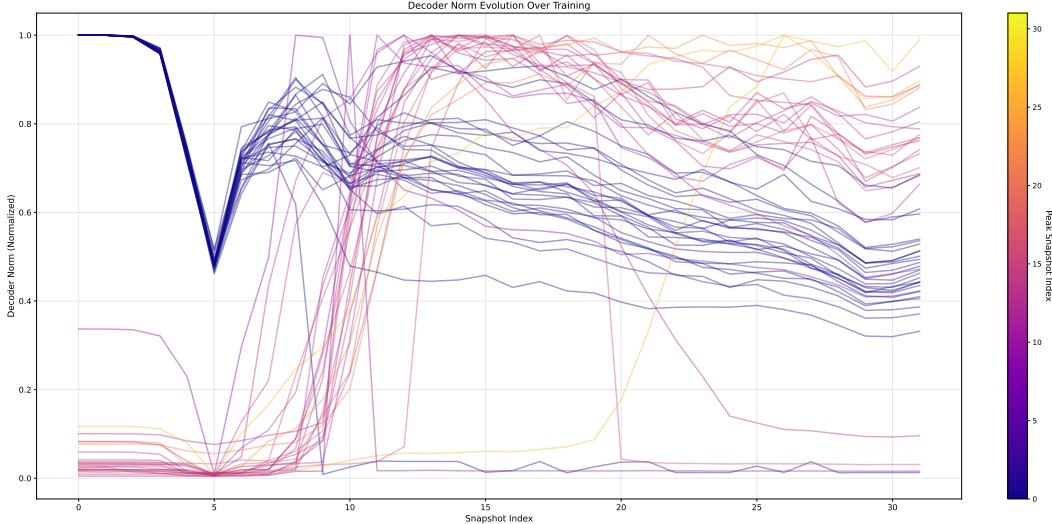

Figure 27: Overview of cross-snapshot feature decoder norm evolution in Pythia-160M Seed2.

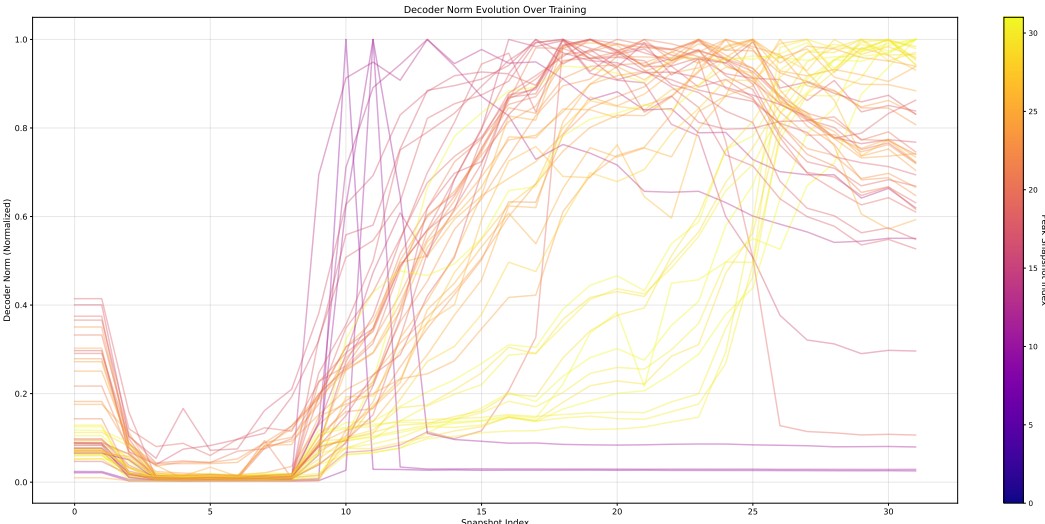

Figure 28: Overview of cross-snapshot feature decoder norm evolution in Alias run of Stanford CRFM GPT-2.

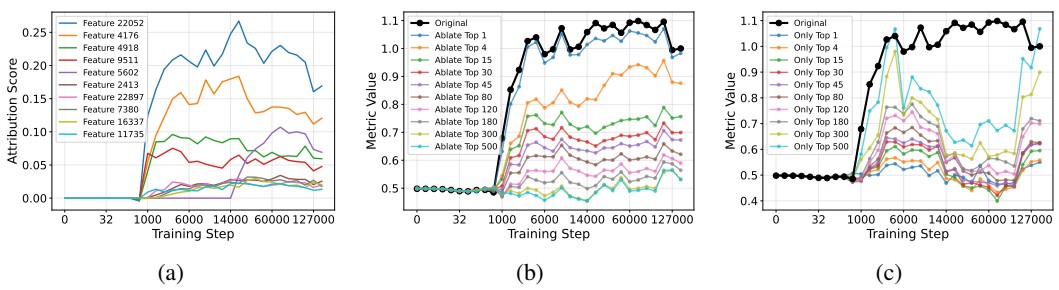

Figure 29: Crosscoder feature attribution on the Across-PP variant of the SVA task, using a crosscoder with 24,576 features trained on Pythia-160M Seed1.

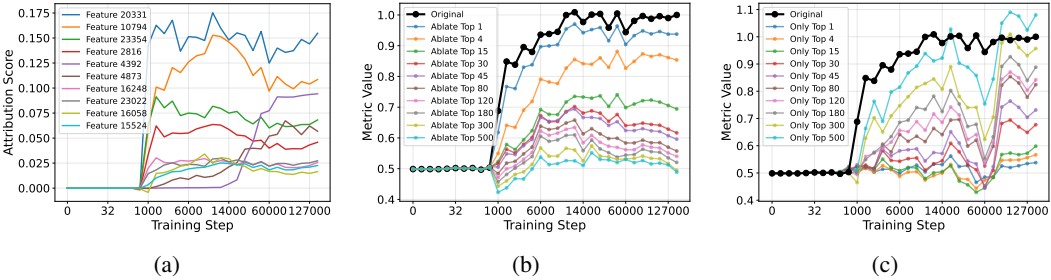

Figure 30: Crosscoder feature attribution on the Across-PP variant of the SVA task, using a crosscoder with 24,576 features trained on Pythia-160M Seed2.

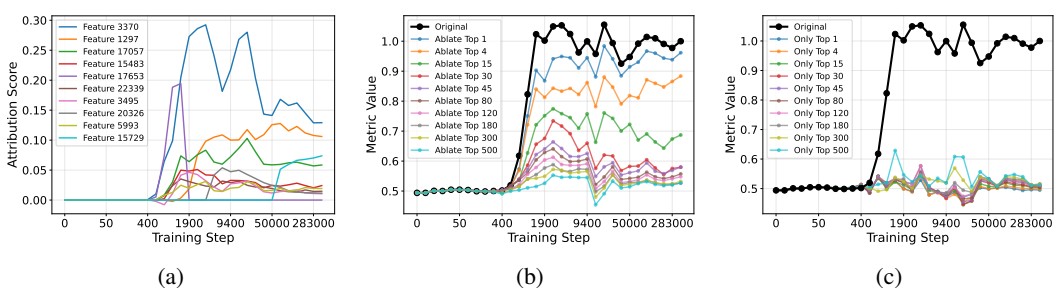

Figure 31: Crosscoder feature attribution on the Across-PP variant of the SVA task, using a crosscoder with 24,576 features trained on the Alias run of Stanford CRFM GPT-2.

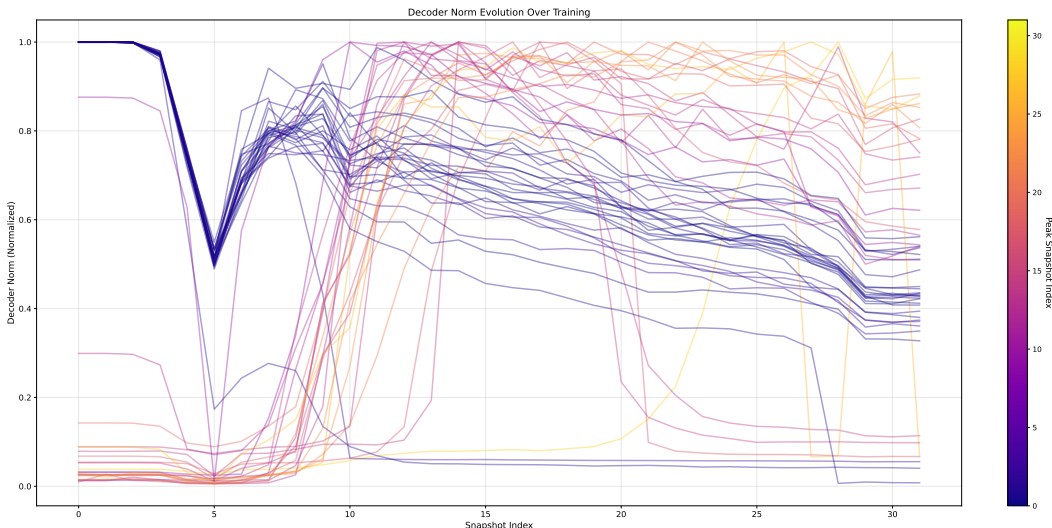

Figure 32: Overview of cross-snapshot feature decoder norm evolution in Pythia-160M, trained on FineWeb-Edu dataset.

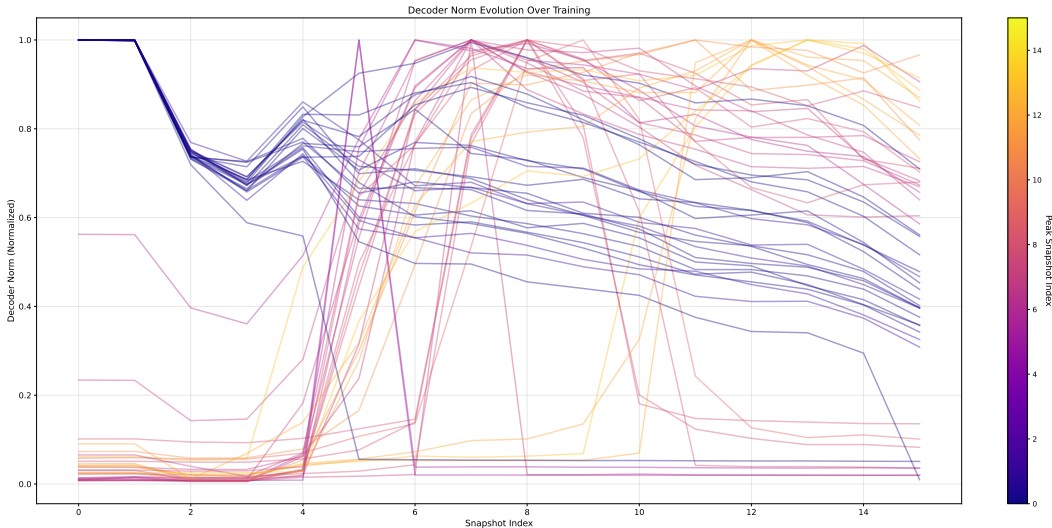

Figure 33: Overview of cross-snapshot feature decoder norm evolution in Pythia-160M, trained on 16 snapshots.

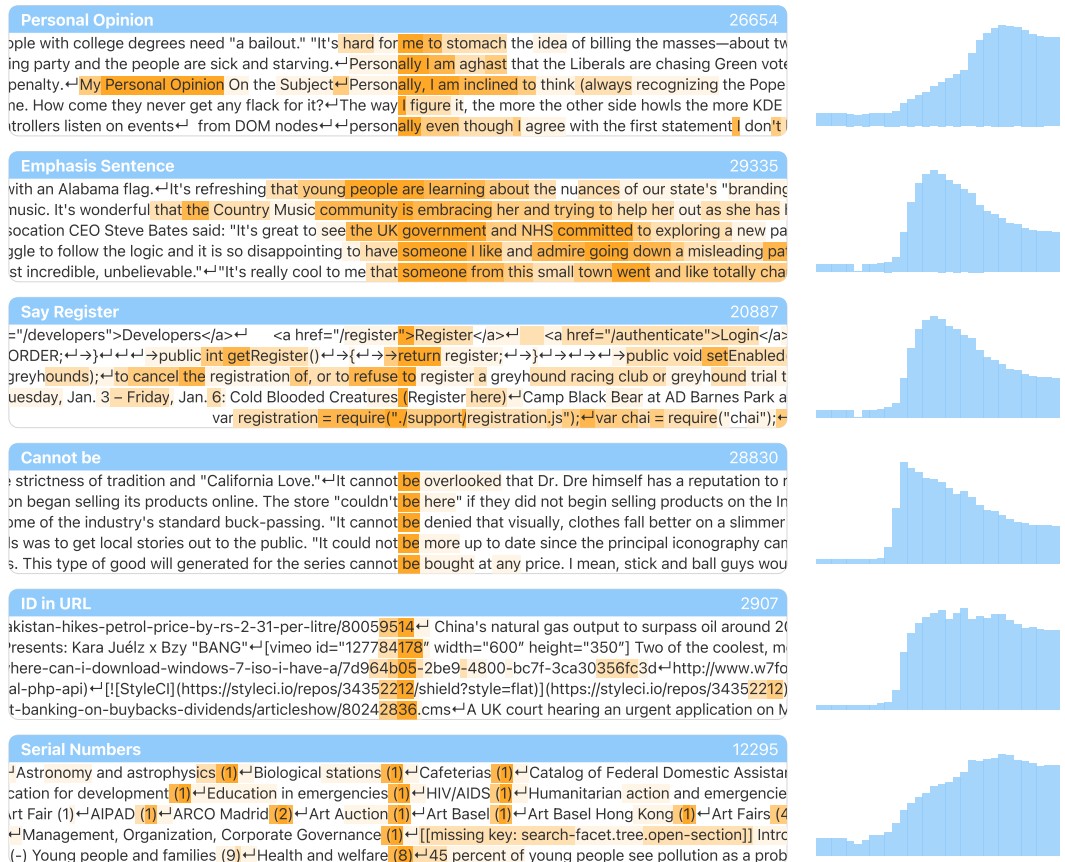

Figure 34: More features in the 32,768-feature crosscoder on Pythia-6.9B

