# OpenReview forum: "Evolution of Concepts in Language Model Pre-Training"
_ICLR.cc/2026/Conference — ICLR 2026 Poster_

### Official Review · Reviewer_effr · 2025-10-17

**Soundness:** 3
**Presentation:** 2
**Contribution:** 3
**Rating:** 4
**Confidence:** 3

**Summary:**

The paper separates the pretraining process of large language models (LLMs) into two distinct phases: a statistical learning phase and a feature learning phase. The authors argue that downstream performance improvements are closely tied to the emergence of features during the latter phase. To quantify this, they introduce a “crosscoder” network that detects features in activations with respect to a so-called decoder norm and track the development of this norm during pretraining. The paper presents empirical results linking these phases to downstream task performance.

**Strengths:**

1) The idea of analyzing the emergence of features during pretraining is interesting and aligns with the current interest in the mechanistic interpretability of LLMs.

2) The paper attempts to link internal feature emergence in the model to downstream generalization, which is a valuable direction.

**Weaknesses:**

1) The decoder norm is part of the training objective, and the weights in the norm are (obviously) part of the model that constructs the feature.
I find this quantity borderline to abstract to serve as a metric that should verify the central concepts of this paper.

2) Figures 4–6 are central to the main claims but are difficult to interpret, as the x-axis is on a logarithmic scale representing training steps.
The authors claim that two distinct learning phases appear, but on a linear scale, this might resemble a single continuous (albeit noisy) learning process. The choice of log-scale seems to visually exaggerate the phase transition; this requires at least discussion and justification. As a remark: on a linear scale, the "statistical phase" is less than 1% of training time.

3) Figure 5 introduces an additional LLM for the interpretation of features, which adds another layer of complexity and potential confounding effects. For me, the interpretation of LLMs should ideally come without LLMs to avoid a circle.

4) Minor remark: Including a figure before the abstract is unconventional.

**Questions:**

1) Why did you choose a log-scale for presenting all results?

2) As all experiments use Pythia, I would like to see a discussion of the training algorithm used there. How many warm-up steps were used? When did the learning rate change?   There is a (small) chance that the seen effects may be trivial if they can be related to, for example, a learning rate change.

3) The y-axis states a "normalized decoder norm". What is normalized exactly? It appears to me that each feature reaches one at some point, which may again make the results less surprising, as each feature must either be present at the start or emerge at some point if one is eventually reached.

---

> ### Author Response · Authors · 2025-11-21
> **Validity of Decoder Norms as Feature-Strength Proxy (1/5)**
>
> > (Reviewer QXzd) Could decoder-norm scaling artifacts or feature-splitting affect the interpretation of “feature emergence”?
>
> > (Reviewer effr) The decoder norm is part of the training objective, and the weights in the norm are (obviously) part of the model that constructs the feature. I find this quantity borderline to abstract to serve as a metric that should verify the central concepts of this paper.
>
> We thank the reviewers for raising concerns about the validity of decoder norms as proxies for feature strength. This property of decoder norms is architectural: if a feature does not exist at a certain snapshot, the sparsity penalty suppresses the feature by pushing its decoder norm toward zero until none of its pre-activations surpass the JumpReLU threshold. Conversely, if a feature is strong at a certain snapshot, this must be reflected in its decoder norm to reduce reconstruction error, since feature activations are shared across snapshots.
>
> Beyond these theoretical considerations, we empirically validate whether decoder norms serve as reliable quantitative indicators of feature evolution in Appendix C. We train linear probing classifiers separately on activations from each model snapshot to predict whether each crosscoder feature activates. The results reveal a strong negative linear relationship between probe errors and crosscoder decoder norms, demonstrating that decoder norms effectively indicate feature evolution.

---

> ### Author Response · Authors · 2025-11-21
> **Scale of X-Axes in Figures (2/5)**
>
> > (Reviewer uCjx) Figure 3c is a bit difficult to interpret, since the analysis references the number of training steps but the figure is labeled in terms of snapshot number. Since there is a stratified sampling approach to selecting snapshots (and since the Pythia models use log-spaced snapshots + snapshots every 1000 training steps), it's not obvious which training step corresponds to which snapshot in the figure and hard to verify the accuracy of the claims.
>
>
> Thanks for pointing out the need for clarification regarding the x-axes in our figures. In our main experiments, we select 32 pretraining snapshots at specific steps for both Pythia 160M and 6.9B, which are exactly: $0,2,4,8,16,32,64,128,256,512,1000,2000,3000,4000,5000,6000,7000,8000,9000,10000$,$14000,20000,27000,34000,47000,60000,74000,87000,100000,114000,127000,143000$. A visualization of how training steps correspond to snapshot indices is provided in Figure 9. The x-axes in Figure 3, 4(b), 4(c), 5, 6 and 7(c) are scaled according to snapshot indices and their corresponding training steps. To facilitate interpretation of how metrics evolve during pretraining, most figures are labeled with training steps rather than snapshot indices. This represents only a relabeling: for example, training step 32 corresponds to snapshot index 5, and training step 1000 corresponds to snapshot index 10.
>
> > (Reviewer effr) Why did you choose a log-scale for presenting all results?
>
> The training steps presented in the aforementioned figures are not uniformly log-spaced, as they correspond to stratified sampling of available training snapshots. We cannot perform fully exponential sampling due to the limited availability of publicly accessible Pythia snapshots. Nevertheless, we find it natural to employ near-exponential sampling with enhanced temporal resolution during the early stages of LM pretraining for the following reasons:
>
> 1. Numerous prior studies on scaling laws have established power-law scaling of loss versus data [1][2][3], indicating the disproportionate impact of early-stage training on loss reduction.
> 2. The cross-entropy loss of both Pythia 160M and 6.9B experiences rapid decline during early training, as shown in Figures 6(a) and 6(b). The loss drops from above 10 to approximately 4 by training step 1000, after which the decline becomes much more gradual.
> While the statistical phase surely constitutes less than 1% of total training time, we find its presence highly significant for the following reasons:
>
> 1. No emergent features form before the turning point at step 1000, despite substantial training loss reduction during early training.
> 2. The feature projections in Figure 4(c) reveal a clear distinction between pre-1000 and post-1000 feature directions. Pre-1000 features exhibit high similarity before step 1000, as do post-1000 features after step 1000. Since this is a pairwise projection analysis, these patterns are not affected by the near-exponential snapshot sampling strategy.
>
> [1] [Scaling Laws for Neural Language Models](https://arxiv.org/abs/2001.08361)
>
> [2] [Training Compute-Optimal Large Language Models](https://arxiv.org/abs/2203.15556)
>
> [3] [Scaling Data-Constrained Language Models](https://arxiv.org/abs/2305.16264)

---

> ### Author Response · Authors · 2025-11-21
> **Learning Rate Scheduling of Pythia (3/5)**
>
> > As all experiments use Pythia, I would like to see a discussion of the training algorithm used there. How many warm-up steps were used? When did the learning rate change? There is a (small) chance that the seen effects may be trivial if they can be related to, for example, a learning rate change.
>
> Thanks for this insightful question. According to the GPT-NeoX configuration in the Pythia paper and their provided [example](https://github.com/EleutherAI/gpt-neox/blob/main/configs/pythia/160M.yml) for reproducing Pythia pretraining, a warmup ratio of 0.01 is used (i.e., 1,430 steps out of 143,000 total steps). We cannot ablate the effect of warmup since virtually all modern transformer training employs warmup, and removing it causes significant performance degradation. However, we do not believe the observed effects are merely artifacts of warmup, based on the following evidence:
>
> 1.  The warmup persists until step 1,430, whereas emergent features begin rising between steps 256 and 1,000, before warmup concludes.
> 2.  Despite the smaller learning rate during warmup, the loss drops rapidly from the beginning of training.
>
> We acknowledge that understanding how the two-stage dynamics interact with learning rate schedules remains an open question. Nevertheless, our proposed method provides a useful tool for inspecting pretraining dynamics, regardless of the underlying causal mechanisms.

---

> ### Author Response · Authors · 2025-11-21
> **Automated Interpretation via LLM (4/5)**
>
> > Figure 5 introduces an additional LLM for the interpretation of features, which adds another layer of complexity and potential confounding effects. For me, the interpretation of LLMs should ideally come without LLMs to avoid a circle.
>
> Thanks for your constructive feedback. We leverage an additional LLM in Figure 5 mainly for a fair, blinded test for how feature complexity correlates to the timing it develops. For the reliability of LLM use in interpreting features (i.e., autointerp), we can provide a brief literature review:
>
> **Positive evidence:**
>
> - (Quantitive) In the [original autointerp paper](https://openaipublic.blob.core.windows.net/neuron-explainer/paper/index.html#sec-human-scoring), OpenAI researchers report an ~0.8 agreement rate on autointerp scores. The experiment was conducted on raw neurons instead of SAE features, so the investigated top activations include both poly- and mono-semantic patterns.
> - (Might be subjective) Anthropic Interp Team leverages autointerp to present their results in their [SAE paper](https://transformer-circuits.pub/2023/monosemantic-features/vis/a1.html) and [circuit tracing paper](https://transformer-circuits.pub/2025/attribution-graphs/biology.html?slug=calc-36-plus-59#french-multilignual-big-svg). These results make intuitive sense for most features they studied, and autointerp scores often align with our intuition.
> - (Might be subjective) Neuronpedia Team provides autointerp for gemma scope and llama scope SAEs, which is often accurate for searching features we are interested in. ([Link](https://www.neuronpedia.org/gemma-2-2b/?sourceSet=gemmascope-mlp-16k&selectedLayers=[]&sortIndexes=[]&ignoreBos=true&q=Chef%20Ramirez%27s%20signature%20dish%2C%20a%20fusion%20of%20Thai%20and%20Mexican%20cuisine%2C%20features%20spicy%20green%20curry%20enchiladas%2C%20tantalizing%20the%20taste%20buds%20with%20a%20bold%2C%20innovative%20flavor%20combination.))
>
> **Negative evidence:**
> - [This ICLR submission](https://openreview.net/forum?id=USyGD0eUod) states that autointerp metrics cannot tell between SAEs trained on randomly initialized transformers and pretrained models.
> - [This work by Anthropic](https://transformer-circuits.pub/2023/monosemantic-features/index.html#global-analysis-about-model) reports similar results that SAEs for untrained transformers contain almost exclusively single-token features (which may due to the fact that for randomly initialized RoPE based transformers, residual stream activations are nearly a function of the current token plus some noise from attention).
>
> Our interpretation of these results is that these single token features contribute a lot to autointerp scores for untrained transformers since they are easier to understand and simulate. So it wont render autointerp completely useless.
>
> Admittedly, automated interpretability is not perfect, but it does provide a labor-free method for rapid feature evaluation. To mitigate the limitations of automated interpretability, we also manually verified the complexity scores generated by the LLM and found that, despite occasional failures, Claude Sonnet 4 correctly recognized most patterns. We therefore consider at least using a stronger LLM to interpret features from a weaker model to be a useful tool when applied with appropriate caution.

---

> ### Author Response · Authors · 2025-11-21
> **Clarification of Normalized Decoder Norm (5/5)**
>
> > The y-axis states a "normalized decoder norm". What is normalized exactly? It appears to me that each feature reaches one at some point, which may again make the results less surprising, as each feature must either be present at the start or emerge at some point if one is eventually reached.
>
> Thanks for your raising concerns regarding normalized decoder norms. The term "normalized" refers to rescaling each feature's decoder norms (across different snapshots) such that the maximum value becomes 1. We follow the original crosscoder paper [1] in applying this rescaling. This naturally ensures that each feature reaches a maximum of 1 at some point during training. The original decoder norms do not carry absolute meaning since they are multiplied by the feature activation $f(x)$ to obtain the actual feature contribution in a given activation. Consequently, decoder norms can exist at arbitrary scales, and only the relative values between decoder norms of the same feature (where $f(x)$ is constant) are meaningful. We apply this rescaling purely for improved visualization.
>
> Nevertheless, the distinction between initialization features and emergent features remains significant. Following the formal definitions, initialization features are defined as:
>
> $$
> \mathcal{F} _ \text{initial}= \\{ i \mid || W _ {\text{dec}, i}^{\theta_0} || > \tau \\}
> $$
>
> and conversely, emergent features as:
>
> $$
> \mathcal{F} _ \text{emergent}= \\{i \mid || W _ {\text{dec}, i}^{\theta_0} || \le \tau \\}
> $$
>
> where $\tau$ is a small threshold chosen to be large enough to eliminate the effect of JumpReLU, since a feature with sufficiently small decoder norm will not activate on any data (see Eq. (8)). We empirically set $\tau = 0.3$ in practice.
>
> These symmetrical definitions partition all features into either initialization or emergent categories, which may appear unremarkable at first glance. However, we observe empirically distinct properties between these two groups:
>
> 1. Initialization features all exhibit peak decoder norms at step 0, while emergent features reach their peaks between step 1,000 (snapshot 10) and step 127,000 (snapshot 30), with most peaking around step 6,000 (snapshot 15). See Figures 3 and 4(a).
> 2. Initialization features correspond to token-level patterns (which is expected given their activation under random initialization), whereas emergent features encode more complex information, including bigrams, phrases, and context-level patterns. See Figure 5 and Figure 24 for examples of emergent features.
>
> For comparison, the decoder norm evolution in cross-layer features can be found here, which illustrates analogous dynamics.
>
> [1] [Sparse Crosscoders for Cross-Layer Features and Model Diffing](https://transformer-circuits.pub/2024/crosscoders/index.html)

---

> > ### Comment · Reviewer_effr · 2025-11-24
> >
> > I thank the authors for their detailed response. However, I will not change my score. One of the paper’s two main contributions, the claim of two distinct learning phases, is demonstrated on only a single model, which makes it impossible to rule out a trivial explanation based on the evidence provided. Such a trivial effect could be something like “once the learning rate is large enough, features begin to emerge,” but it might also be something entirely different that none of us has considered yet. In my view, to support a claim of this kind, an analysis of at least three models would be necessary.

---

> ### Author Response · Authors · 2025-11-24
> **Update on Generalization Results on Different Models**
>
> We understand the remaining concerns you have on generalizability of our findings. We would like to share additional results we have not provided in our previous comments.
>
> - **Different initializations (Pythia 160M variants):** We observe highly consistent results across differently initialized versions of Pythia 160M. Decoder norms of initialization features follow identical trajectories, and emergent features begin rising around step 1000, exhibiting a clear two-phase pattern with the same transition point (Figure 25 and 26 of the revised manuscript). Attribution experiments confirm that we successfully identify sparse crosscoder features with significant contributions to downstream tasks. We show these results in Figure 28 and 29 of the revised manuscript.
>
> - **Different base models (GPT-2):** For Stanford CRFM's GPT-2 model, we observe a clear two-phase pattern, with emergent features rising around step 1,000. Upon manual inspection of these features, we still find that initialization features exhibit token-level information, while more complex patterns emerge in emergent features.
>
>   However, we find a notable difference: due to absolute positional embeddings, initialization feature norms do not peak at the beginning but rather at later steps (with above-threshold norm at beginning). This difference arises because the absolute positional embeddings initially exhibit large norms (larger than the word embeddings) and dominate the activation norm. Since we normalize all activation sources to the same average activation norm $\sqrt{d_\text{model}}$, this large positional contribution substantially reduces the observed norms of initialization features. We show these results in Figure 27 of the revised manuscript.
>
>   To sum up, the statistical-to-feature-learning phase transition still exists across models, despite small differences. Attribution experiments (Figure 30 of the revised manuscript) again confirm the effectiveness of sparse crosscoder features, with generally similar feature evolution patterns for specific downstream tasks.
>
> - **Other models with accessible checkpoints:** We also investigated other model families, including [BLOOM](https://arxiv.org/abs/2211.05100) and [OLMo](https://github.com/allenai/OLMo). Unfortunately, these models either lack sufficient intermediate checkpoints for fine-grained analysis of pretraining dynamics (BLOOM), or lack sufficiently early checkpoints to capture the two-stage transition (BLOOM and OLMo).
>
> We include all these findings above in Appendix G.1. We believe there is still room for more consolidated experiments supporting our claim. But due to limitations of existing resources for pretraining & scaling research, we are unable to go further in this direction for now. Nonetheless, our current experiments on 3 seeds of Pythia-160M (original Pythia suite + 2 PolyPythia seeds), 2 sizes of Pythia models (160M & 6.9B) and 2 model families (Pythia + Stanford CRFM GPT2) can serve as strong evidence.
>
> Even if we have tried our best effort to conduct our experiments on all pretraining checkpoints available, we still cannot rule out the possibility of the learning rate hypothesis since it is unconventional to train language models without learning rate warmup. Stanford CRFM GPT2 models also use [lr warmup](https://github.com/stanford-crfm/mistral/blob/main/conf/trainers/gpt2-small.yaml#L43). We thank you for your valuable suggestions and we politely ask on your reconsideration on the reliability of our claims.
>
> [1] [PolyPythias: Stability and Outliers across Fifty Language Model Pre-Training Runs](https://arxiv.org/abs/2503.09543)
>
> [2] https://github.com/stanford-crfm/mistral

---

> > ### Comment · Reviewer_effr · 2025-11-28
> >
> > Having read the appendix, I would like to thank the authors for providing additional information. However, I do not change my score. The analysis is based on plots that lack a clear mathematical framework for interpretation.  This would be fine in principle if the setup were broad enough, but I do not believe that the current empirical validation is broad enough.

---

### Official Review · Reviewer_QXzd · 2025-10-29

**Soundness:** 3
**Presentation:** 2
**Contribution:** 2
**Rating:** 6
**Confidence:** 1

**Summary:**

This paper investigates how internal features in large language models emerge and evolve during pre-training. Using cross-snapshot crosscoders—a variant of sparse autoencoders adapted to align activations across training checkpoints—the authors track interpretable features throughout the training process of Pythia models. They identify initialization and emergent features, quantify their persistence, and show that feature complexity increases with training progress. Attribution-based analyses link these microscopic feature dynamics to downstream task improvements. The study further reveals a clear two-phase transition in pre-training, from an early statistical learning phase dominated by token-level regularities to a later feature learning phase characterized by sparse, semantically rich representations.

**Strengths:**

The paper presents a technically novel adaptation of crosscoders for temporal analysis, offering the first fine-grained view of feature evolution across pre-training snapshots. The experiments are extensive and carefully controlled, combining mechanistic interpretability with learning dynamics. The empirical discovery of a statistical-to-feature-learning transition provides a coherent mechanistic complement to theoretical frameworks such as the information bottleneck and singular learning theory. The work stands out for its methodological clarity, strong empirical validation, and relevance to both interpretability and representation learning research.

**Weaknesses:**

The study provides interesting descriptive evidence, but the causal claims remain largely correlational—attribution patching is still a heuristic rather than proof of necessity.
The analysis is limited to Pythia checkpoints and relatively simple syntactic tasks, so generalization to larger or more complex models is unclear.
Some methodological choices (e.g., decoder-norm as feature-strength proxy, selection of snapshots) lack formal justification or ablation.
The paper is also quite engineering-heavy, and clarity could improve in conveying what new conceptual insight the cross-coder adds beyond existing SAE frameworks.

**Questions:**

How sensitive are the results to the number or spacing of snapshots?

Could decoder-norm scaling artifacts or feature-splitting affect the interpretation of “feature emergence”?

Does attribution patching preserve causal validity when features are highly correlated?

Have the authors verified that the same cross-coder features generalize across layers or different datasets?

---

> ### Author Response · Authors · 2025-11-21
> **Sensitivity of Snapshot Selection (1/5)**
>
> > How sensitive are the results to the number or spacing of snapshots?
>
> Thanks for your constructive question. Our main experiments use crosscoders trained on 32 pretraining snapshots from each model. To test the sensitivity of our methods to snapshot selection, we trained an additional crosscoder on Pythia 160M using 16 pretraining snapshots, downsampled from the original set by selecting every other snapshot (`steps=steps[::2]`). We find that the overall feature evolution exhibits nearly identical trends, with only a reduction in temporal resolution. Detailed results have been incorporated in Appendix G.3.

---

> ### Author Response · Authors · 2025-11-21
> **Validity of Decoder Norms as Feature-Strength Proxy (2/5)**
>
> > (Reviewer QXzd) Could decoder-norm scaling artifacts or feature-splitting affect the interpretation of “feature emergence”?
>
> > (Reviewer effr) The decoder norm is part of the training objective, and the weights in the norm are (obviously) part of the model that constructs the feature. I find this quantity borderline to abstract to serve as a metric that should verify the central concepts of this paper.
>
> We thank the reviewers for raising concerns about the validity of decoder norms as proxies for feature strength. This property of decoder norms is architectural: if a feature does not exist at a certain snapshot, the sparsity penalty suppresses the feature by pushing its decoder norm toward zero until none of its pre-activations surpass the JumpReLU threshold. Conversely, if a feature is strong at a certain snapshot, this must be reflected in its decoder norm to reduce reconstruction error, since feature activations are shared across snapshots.
>
> Beyond these theoretical considerations, we empirically validate whether decoder norms serve as reliable quantitative indicators of feature evolution in Appendix C. We train linear probing classifiers separately on activations from each model snapshot to predict whether each crosscoder feature activates. The results reveal a strong negative linear relationship between probe errors and crosscoder decoder norms, demonstrating that decoder norms effectively indicate feature evolution.

---

> ### Author Response · Authors · 2025-11-21
> **Validity of Attribution Patching (3/5)**
>
> > Does attribution patching preserve causal validity when features are highly correlated?
>
> Thanks for your question. Attribution patching is a currently generally used method for isolating critical internal components, used both at head level [1][2] and sparse feature level [3][4]. This work adopts the idea of attribution patching and converts it to crosscoder case to consistently capture critical features and their evolution. As far as I know, there's no previous research on the causal validity of attribution patching when features are highly correlated, and this is a bit out of scope for our work. But we think it's likely the causal validity persists in correlated features for the little feature-wise interaction. And the ablation study of faithfulness and completeness should tell the failure (if exists).
>
> [1] [Attribution Patching: Activation Patching At Industrial Scale](https://www.neelnanda.io/mechanistic-interpretability/attribution-patching)
>
> [2] [Attribution Patching Outperforms Automated Circuit Discovery](https://arxiv.org/abs/2310.10348)
>
> [3] [Automatically identifying local and global circuits with linear computation graphs](https://arxiv.org/abs/2405.13868)
>
> [4] [Sparse feature circuits: Discovering and editing interpretable causal graphs in language models. In The Thirteenth International Conference on Learning Representations](https://openreview.net/forum?id=I4e82CIDxv)

---

> ### Author Response · Authors · 2025-11-21
> **Generalization across Datasets (4/5)**
>
> > (Reviewer zNuD) Could the authors clarify the rationale behind choosing SlimPajama as the pretraining dataset? Were other datasets considered, and if so, why were they excluded?
>
> > (Reviewer QXzd) Have the authors verified that the same cross-coder features generalize across layers or different datasets?
>
> We appreciate the reviewers' concerns regarding the generalizability of our methods and analyses across different datasets. SlimPajama is a widely used deduplicated text corpus for pretraining that has been employed in prior work on sparse dictionary learning [1]. We selected SlimPajama primarily because it is a standard corpus with sufficient data volume for our experiments.
>
> To further demonstrate the robustness of our methods and analyses across datasets, we trained crosscoders for Pythia 160M on the FineWeb-Edu dataset [2], which consists of educational web pages. Our results show nearly identical patterns to those observed with Pythia 160M crosscoders trained on SlimPajama, exhibiting very similar feature evolution dynamics. We have incorporated detailed results from these experiments in Appendix G.2.
>
> [1] [Llama Scope: Extracting Millions of Features from Llama-3.1-8B with Sparse Autoencoders](https://arxiv.org/abs/2410.20526)
>
> [2] [The FineWeb Datasets: Decanting the Web for the Finest Text Data at Scale](https://arxiv.org/abs/2406.17557)

---

> ### Author Response · Authors · 2025-11-21
> **Generalization across Layers (5/5)**
>
> > Have the authors verified that the same cross-coder features generalize across layers or different datasets?
>
> For the generalization across layers, we have included results on all layers of Pythia 160M. We find most middle layers exhibit the same evolutionary patterns as Layer 6, supporting our findings in Section 4.

---

### Official Review · Reviewer_uCjx · 2025-10-31

**Soundness:** 4
**Presentation:** 4
**Contribution:** 3
**Rating:** 8
**Confidence:** 4

**Summary:**

The authors study the evolution of features during LLM training using crosscoder SAEs. They first study the emergence, persistence of features in LLMs, showing that many features persist for long periods and that their emergence time varies. They then consider feature complexity, showing a moderate correlation between feature complexity and the time at which the feature's emergence peaks. After analyzing feature dynamics, they study the causal relationship between features and task performance, using attribution patching to assess causal relationships for subject-verb agreement, indirect object identification, and induction tasks. Finally, they show that the initial part of training learns statistical features of token distributions and study the changes in feature dimensionality during training.

**Strengths:**

The authors introduce a novel crosscoder method for studying feature dynamics in LLMs. They analysis of feature emergence and causal study is clear and thorough.

**Weaknesses:**

Figure 3c is a bit difficult to interpret, since the analysis references the number of training steps but the figure is labeled in terms of snapshot number. Since there is a stratified sampling approach to selecting snapshots (and since the Pythia models use log-spaced snapshots + snapshots every 1000 training steps), it's not obvious which training step corresponds to which snapshot in the figure and hard to verify the accuracy of the claims.

The argument about feature dimensionality and compression is a bit unclear with an unclear takeaway.

Only relying on one family of models reduces the potential impact.

**Questions:**

How do the results in 6b and 6c compare with a similar plot produced by ablating all but/ablating $k$ random features?

Could you produce similar analysis with additional models with available checkpoints, such as Stanford CRFM GPT-2 or Olmo?

---

> ### Author Response · Authors · 2025-11-21
> **Scale of X-Axes in Figures (1/4)**
>
> > (Reviewer uCjx) Figure 3c is a bit difficult to interpret, since the analysis references the number of training steps but the figure is labeled in terms of snapshot number. Since there is a stratified sampling approach to selecting snapshots (and since the Pythia models use log-spaced snapshots + snapshots every 1000 training steps), it's not obvious which training step corresponds to which snapshot in the figure and hard to verify the accuracy of the claims.
>
>
> Thanks for pointing out the need for clarification regarding the x-axes in our figures. In our main experiments, we select 32 pretraining snapshots at specific steps for both Pythia 160M and 6.9B, which are exactly: $0,2,4,8,16,32,64,128,256,512,1000,2000,3000,4000,5000,6000,7000,8000,9000,10000$,$14000,20000,27000,34000,47000,60000,74000,87000,100000,114000,127000,143000$. A visualization of how training steps correspond to snapshot indices is provided in Figure 9. The x-axes in Figure 3, 4(b), 4(c), 5, 6 and 7(c) are scaled according to snapshot indices and their corresponding training steps. To facilitate interpretation of how metrics evolve during pretraining, most figures are labeled with training steps rather than snapshot indices. This represents only a relabeling: for example, training step 32 corresponds to snapshot index 5, and training step 1000 corresponds to snapshot index 10.
>
> > (Reviewer effr) Why did you choose a log-scale for presenting all results?
>
> The training steps presented in the aforementioned figures are not uniformly log-spaced, as they correspond to stratified sampling of available training snapshots. We cannot perform fully exponential sampling due to the limited availability of publicly accessible Pythia snapshots. Nevertheless, we find it natural to employ near-exponential sampling with enhanced temporal resolution during the early stages of LM pretraining for the following reasons:
>
> 1. Numerous prior studies on scaling laws have established power-law scaling of loss versus data [1][2][3], indicating the disproportionate impact of early-stage training on loss reduction.
> 2. The cross-entropy loss of both Pythia 160M and 6.9B experiences rapid decline during early training, as shown in Figures 6(a) and 6(b). The loss drops from above 10 to approximately 4 by training step 1000, after which the decline becomes much more gradual.
> While the statistical phase surely constitutes less than 1% of total training time, we find its presence highly significant for the following reasons:
>
> 1. No emergent features form before the turning point at step 1000, despite substantial training loss reduction during early training.
> 2. The feature projections in Figure 4(c) reveal a clear distinction between pre-1000 and post-1000 feature directions. Pre-1000 features exhibit high similarity before step 1000, as do post-1000 features after step 1000. Since this is a pairwise projection analysis, these patterns are not affected by the near-exponential snapshot sampling strategy.
>
> [1] [Scaling Laws for Neural Language Models](https://arxiv.org/abs/2001.08361)
>
> [2] [Training Compute-Optimal Large Language Models](https://arxiv.org/abs/2203.15556)
>
> [3] [Scaling Data-Constrained Language Models](https://arxiv.org/abs/2305.16264)

---

> ### Author Response · Authors · 2025-11-21
> **Clarification of Feature Dimensionalites (2/4)**
>
> > The argument about feature dimensionality and compression is a bit unclear with an unclear takeaway.
>
> Thanks for this feedback on the clarity of our feature dimensionality statements. Our feature dimensionality definition follows Elhage et al. (2022) [1], which measures what "fraction of a dimension" a specific feature occupies. The numerator represents the extent to which a given feature is represented, while the denominator captures "how many features share the dimension it is embedded in" by projecting each feature onto its dimension.
>
> It can be proven that the sum of feature dimensionalities across all features is at most equal to the dimension of the activation space. Intuitively, when features are "packed efficiently," these feature dimensionalities sum to the maximum possible value. For example, consider 3 feature vectors embedded in 2 dimensions: when these vectors are symmetrically arranged (forming an equilateral triangle), each feature achieves a dimensionality of $\frac{2}{3}$, summing to $2$ (the activation space dimension). In contrast, a suboptimal placement can be feature vectors at $(0,1)$, $(1,0)$, and $(-\frac{\sqrt{2}}{2},-\frac{\sqrt{2}}{2})$, which yields feature dimensionalities of $\frac{2}{3}$, $\frac{2}{3}$, and $\frac{1}{2}$ respectively, summing to $\frac{11}{6} < 2$.
>
> In Figure 7(c), we can observe that for both models, the total feature dimensionality first decreases and then increases. This pattern reveals that superposition does not simply increase monotonically as the model learns more features. Instead, the model undergoes an internal compression before accommodating new features. While the mechanistic understanding of superposition formation remains an open question, we interpret this phenomenon as evidence of a fitting-to-compression phase transition, consistent with predictions from information bottleneck theory.
>
> [1] [Toy Models of Superposition](https://transformer-circuits.pub/2022/toy_model/index.html#geometry-dimensionality)

---

> ### Author Response · Authors · 2025-11-21
> **Random Baseline of Attribution Results (3/4)**
>
> > How do the results in 6b and 6c compare with a similar plot produced by ablating all but/ablating random features?
>
> Thanks for your insightful question. We have incorporated a random baseline on the Across-PP variant of the SVA task in Appendix E.5. The results show ablating random features hardly affect the original model since the sparse activating nature of crosscoder features.

---

> ### Author Response · Authors · 2025-11-21
> **Generalization across Model Family/Seed (4/4)**
>
> > (Reviewer zNuD) How stable are the observed feature dynamics across different base models or seeds? Would larger or differently initialized models exhibit the same “turning point” behavior?
>
> > (Reviewer uCjx) Could you produce similar analysis with additional models with available checkpoints, such as Stanford CRFM GPT-2 or Olmo?
>
> We appreciate the reviewers' concerns regarding the generalizability of our methods and analyses across different model families and initializations. In addition to our original results on Pythia 160M and Pythia 6.9B (which demonstrate generalizability across model sizes), we conducted experiments on two additional Pythia 160M models with different random seeds (Pythia 160M Seed1 and Seed2) [1] and the Alias run of Stanford CRFM's GPT-2 model [2].
>
> - **Different initializations (Pythia 160M variants):** We observe highly consistent results across differently initialized versions of Pythia 160M. Decoder norms of initialization features follow identical trajectories, and emergent features begin rising around step 1000, exhibiting a clear two-phase pattern with the same transition point (Figure 25 and 26 of the revised manuscript). Attribution experiments confirm that we successfully identify sparse crosscoder features with significant contributions to downstream tasks. We show these results in Figure 28 and 29 of the revised manuscript.
>
> - **Different base model (GPT-2):** For Stanford CRFM's GPT-2 model, we observe a clear two-phase pattern, with emergent features rising around step 1,000. Upon manual inspection of these features, we still find that initialization features exhibit token-level information, while more complex patterns emerge in emergent features.
>
>   However, we find a notable difference: due to absolute positional embeddings, initialization feature norms do not peak at the beginning but rather at later steps (with above-threshold norm at beginning). This difference arises because the absolute positional embeddings initially exhibit large norms (larger than the word embeddings) and dominate the activation norm. Since we normalize all activation sources to the same average activation norm $\sqrt{d_\text{model}}$, this large positional contribution substantially reduces the observed norms of initialization features. We show these results in Figure 27 of the revised manuscript.
>
>   To sum up, the statistical-to-feature-learning phase transition still exists across models, despite small differences. Attribution experiments (Figure 30 of the revised manuscript) again confirm the effectiveness of sparse crosscoder features, with generally similar feature evolution patterns for specific downstream tasks.
>
> - **Other models with accessible checkpoints:** We also investigated other model families, including [BLOOM](https://arxiv.org/abs/2211.05100) and [OLMo](https://github.com/allenai/OLMo). Unfortunately, these models either lack sufficient intermediate checkpoints for fine-grained analysis of pretraining dynamics (BLOOM), or lack sufficiently early checkpoints to capture the two-stage transition (BLOOM and OLMo).
>
> We include all these findings above in Appendix G.1. We believe there is still room for more consolidated experiments supporting our claim. But due to limitations of existing resources for pretraining & scaling research, we are unable to go further in this direction for now. Nonetheless, our current experiments on 3 seeds of Pythia-160M (original Pythia suite + 2 PolyPythia seeds), 2 sizes of Pythia models (160M & 6.9B) and 2 model families (Pythia + stanford GPT2) can serve as strong evidence.
>
> [1] [PolyPythias: Stability and Outliers across Fifty Language Model Pre-Training Runs](https://arxiv.org/abs/2503.09543)
>
> [2] https://github.com/stanford-crfm/mistral

---

### Official Review · Reviewer_zNuD · 2025-11-01

**Soundness:** 1
**Presentation:** 1
**Contribution:** 1
**Rating:** 2
**Confidence:** 4

**Summary:**

The paper claims to explore how meaningful features emerge as language models learn.  Using crosscoders to align representations across training snapshots, the authors trace training dynamics over time. They find a clear turning point early in pretraining, where models move from simple statistical patterns to more structured, feature-based understanding. Through attribution and patching analyses, these evolving features are linked to linguistic behaviors such as agreement and induction.

**Strengths:**

- The paper touches on an interesting and relevant question, how internal features evolve during language model pretraining.

- It attempts to provide a structured view using crosscoders.

- The motivation to study model dynamics rather than static checkpoints is reasonable and aligns with growing interest in understanding training trajectories.

- The work could serve as a preliminary exploration toward more rigorous analyses of concept formation in neural networks.

**Weaknesses:**

- Novelty. The paper offers little substantive novelty beyond prior work on crosscoders (Lindsay et al., 2024). The method follows existing frameworks closely, with minimal theoretical or heuristic grounding for its design choices. For instance,
    - 1) The selection of the SlimPajama dataset is not clearly justified, nor is the limited set of base models sufficient to generalize the findings.
    - 2) An ablation study on activation aggregation in Eq. (1) would help clarify whether summing activations is the optimal or even appropriate choice.

- Contribution. The contribution remains narrow, more a descriptive report than a step toward formal interpretability. The reliance on visualizations such as heatmaps and curves, without clear quantitative or linguistic interpretation, makes it hard to connect the observed patterns to interpretable model behaviors.

- Rigorousness. Several findings, particularly those regarding early n-gram learning, largely lack of contextualization with known results from prior work (e.g., Svete & Cotterell, 2024; Nguyen, 2024, Chen, 2024).

- Writing. The writing quality is also uneven. The paper reads more like a lab report than a cohesive scientific paper. For instance
    - 1) Lack of grounded formalization for key concepts. For example, can we define "initial features" and "emergent features" (line 202-line207) formally using the notations in the paper?
    - 2) Some phrasing is imprecise. "the pre-training process remains largely a black box." The sentence is a bit unclear. The process of pretraining is generally gradient descent on large-scale of datasets. I guess what the authors mean here is "the pretraining dynamic" instead of "process".

Overall, the work lacks the theoretical grounding, novelty, and clarity needed to make a strong contribution. That said, it is definitely worthwhile for presenting at a workshop after revision.


``References``
- Anej Svete and Ryan Cotterell. Transformers can represent n-gram language models. arXiv preprint
arXiv:2404.14994, 2024.
- Timothy Nguyen. Understanding transformers via n-gram statistics. arXiv preprint
arXiv:2407.12034, 2024.
- Chen et al. Jet expansions of residual computation  arXiv preprint 2410.06024, 2024.

**Questions:**

1. Could the authors clarify the rationale behind choosing SlimPajama as the pretraining dataset? Were other datasets considered, and if so, why were they excluded?
2. In Eq. (1), how were activations aggregated across snapshots? Why was summing chosen, and how might alternative strategies (e.g., averaging, normalization, or selective sampling) affect the results?
3. Can “initial features” and “emergent features” be defined more formally within the paper’s notation? A clearer definition would make the results more interpretable and reproducible.
4. Have the authors compared their findings with recent analyses of early-stage n-gram learning (e.g., Svete & Cotterell, 2024; Nguyen, 2024, Chen et al, 2024)? What new insights, if any, extend beyond those studies?
5. How stable are the observed feature dynamics across different base models or seeds? Would larger or differently initialized models exhibit the same “turning point” behavior?

---

> ### Author Response · Authors · 2025-11-21
> **Novelty & Contribution Claim (1/6)**
>
> > Novelty. The paper offers little substantive novelty beyond prior work on crosscoders (Lindsay et al., 2024). The method follows existing frameworks closely, with minimal theoretical or heuristic grounding for its design choices.
>
> > Contribution. The contribution remains narrow, more a descriptive report than a step toward formal interpretability. The reliance on visualizations such as heatmaps and curves, without clear quantitative or linguistic interpretation, makes it hard to connect the observed patterns to interpretable model behaviors.
>
> We appreciate the concerns raised regarding the novelty and contribution of our paper. We would like to clarify that the fundamental research question of our work focus on the **microscopic (feature-level) study of pre-training dynamics**. While the crosscoder architecture mainly follows Lindsay et al., 2024 with per-activation-source encoders and decoders and a summing aggregation in pre-feature-activations, our investigation of pre-training dynamics represents a distinct contribution beyond the original application in studying cross-layer superposition. The main novelty lies in our analyses specific to pre-training dynamics:
>
> 1. We study how microscopic feature evolution is related to changes in macroscopic behavioral changes in model inference, which is a causal effect.
> 2. We quantify the relevance between feature complexity and the timing of its formation with automated interpretability, finding results consistent with our intuition.
> 3. We examine how superposition emerges in the pre-training of language models from a feature-centric perspective.
> 4. We explore potential failure modes of crosscoder in studying feature alignment in Appendix A.6.
>
> To address the distinct challenges in training cross-snapshot crosscoders, we introduce architectural and engineering improvements in Appendices A.1 and A.3 that ensure sparsity, reconstruction fidelity, and training feasibility. Our design choices are grounded in both heuristic reasoning and empirical validation: the architectural discussion in Section 3 (model, snapshot, and dataset selection) and Appendix A.1 (activation and regularization function selection), along with the sparsity-reconstruction comparisons in Appendix A.5 and the validation of crosscoder decoder norms as quantitative indicators of feature evolution in Appendix C.
>
> We take advantage of heatmaps and curves to provide clear, easy-to-follow conclusions. We can easily find the distinction of initialization features and emergent features, suggesting a two-stage pretraining, by observing these heatmaps and curves in Figure 3 and Figure 4. These figures also provide quantitative analyses: all subfigures in Figures 4, 5, and 6 include clearly labeled x/y axes, with statistical significance metrics (Pearson correlation coefficients) in Figure 5 and sufficiency and completeness scores in Figure 6. Regarding linguistic interpretation, we offer detailed explanations of how features emerge and alternate in dominance within the Across-PP variant of the SVA task. Additional examples of crosscoder features and their decoder norm evolution are provided in Appendix H.

---

> ### Author Response · Authors · 2025-11-21
> **Previous Results of Statistical Phase (2/6)**
>
> > Rigorousness. Several findings, particularly those regarding early n-gram learning, largely lack of contextualization with known results from prior work (e.g., Svete & Cotterell, 2024; Nguyen, 2024, Chen, 2024).
>
> > Have the authors compared their findings with recent analyses of early-stage n-gram learning (e.g., Svete & Cotterell, 2024; Nguyen, 2024, Chen et al, 2024)? What new insights, if any, extend beyond those studies?
>
> Thanks for your highlighting relevant prior works on early n-gram learning! We've incorporated these works into our paper. As in Section 6, we have included several related work on models learning statistical patterns during early training stages, including:
>
> 1. Takahashi & Tanaka-Ishii, 2017 [1] demonstrate that LSTM language model can effectively reproduce Zipf's law and Heaps' law, two representative statistical properties underlying natural language;
> 2. Xu et al., 2019 [2] study DNN training behavior in frequency domain, finding that a DNN with common settings first quickly captures the dominant low-frequency components, and then relatively slowly captures the high-frequency ones.
> 3. Choshen et al. 2022 [3] show that neural language models acquire linguistic phenomena in a similar order across architecture and data, in which preceding words and n-grams play a predominant role.
> 4. Belrose et al. 2024 [4] show that neural networks (including Pythia) automatically learn low-order statistics matching those of the training set early in training.
>
> Our work complement the prior works in the following aspects:
>
> 1. We focus on unigram and bigram behavior of transformers (achievable without the attention mechanism), and find the language model model almost exclusively learns these statistical behavior by checking the uni- and bi-gram entropy.
> 2. We characterize the transition from statistical learning to feature learning through the lens of superposition, observing that total feature dimensionality first decreases then increases, marking a distinct phase change.
>
> [1] [Do neural nets learn statistical laws behind natural language?](http://arxiv.org/abs/1707.04848)
>
> [2] [Training behavior of deep neural network in frequency domain](https://arxiv.org/abs/1807.01251)
>
> [3] [The grammar-learning trajectories of neural language models](https://arxiv.org/abs/2109.06096)
>
> [4] [Neural networks learn statistics of increasing complexity](https://openreview.net/forum?id=IGdpKP0N6w)

---

> ### Author Response · Authors · 2025-11-21
> **Aggregation of Pre-Feature-Activations (3/6)**
>
> > In Eq. (1), how were activations aggregated across snapshots? Why was summing chosen, and how might alternative strategies (e.g., averaging, normalization, or selective sampling) affect the results?
>
> Our aggregation strategy for pre-feature-activations follows the acausal variant of Lindsay et al. (2024), which directly sums per-snapshot pre-feature-activations. Alternative strategies such as averaging should preserve the same representational capacity. For instance, feature activation through averaging per-snapshot pre-feature-activations
>
> $$
> f'(x)=\sigma\left(\frac{1}{|\Theta|}\sum_{\theta \in \Theta} W_{\text {enc}}^\theta a^\theta(x)+b_{\text {enc}}\right)
> $$
>
> can be equivalently represented by learning ${W^\theta_\text{enc}}' = |\Theta| W^\theta_\text{enc}$, a simple rescaling of the original encoder weights.
> Lindsay et al. (2024) propose other crosscoder variants, including weakly/strongly causal, local, and skip-layer crosscoders, each with different aggregation strategies. These strategies impose extra additional inductive bias on different activation sources (i.e. activations from different snapshots in our work) and break their symmetry, which may lead to features formed under these prior biases.
>
> More complex strategies like normalization and selective sampling introduce extra non-linearity and complexity. Since we have validatged the effectiveness of crosscoders under the simpler summing strategy (Appendices A, B and C), there is limited motivation to adopt more complex aggregation functions, following the principle of Occam's razor. Furthermore, these non-linearities would complicate future circuit analyses [1][2] of crosscoder features. Given that the aggregation of pre-feature-activations is just intended to create a unified feature activation allowing all decoder vectors to respond to all encoders, these alternative strategies would likely yield similar results for detecting feature evolution.
>
> [1] [Automatically identifying local and global circuits with linear computation graphs](https://arxiv.org/abs/2405.13868)
>
> [2] [Sparse feature circuits: Discovering and editing interpretable causal graphs in language models. In The Thirteenth International Conference on Learning Representations](https://openreview.net/forum?id=I4e82CIDxv)

---

> ### Author Response · Authors · 2025-11-21
> **Definition of Initialization Features and Emergent Features (4/6)**
>
> > Lack of grounded formalization for key concepts. For example, can we define "initial features" and "emergent features" (line 202-line207) formally using the notations in the paper?
>
> > Can “initial features” and “emergent features” be defined more formally within the paper’s notation? A clearer definition would make the results more interpretable and reproducible.
>
> Thanks for pointing this out. From a perspective of formal definition, the distinction of initialization features and emergent features is straightforward: initialization features exist at beginning, while emergent features do not. Accordingly, we can define the set of initialization features formally as:
>
> $$
> \mathcal{F} _ \text{initial}= \\{ i \mid || W _ {\\text{dec}, i}^{\\theta_0} || > \tau \\}
> $$
>
> and conversely, emergent features as:
>
> $$
> \mathcal{F} _ \text{emergent}= \\{i \mid || W _ {\\text{dec}, i}^{\\theta_0} || \le \tau \\}
> $$
>
> where $\tau$ is a small threshold chosen to be large enough to eliminate the effect of JumpReLU, since a feature with sufficiently small decoder norm will not activate on any data (see Eq. (8)). We empirically set $\tau = 0.3$ in practice.
>
> These symmetrical definitions partition all features into either initialization or emergent categories, which may appear unremarkable at first glance. However, we do observe empirically distinct properties between these two groups:
>
> 1. Initialization features all exhibit peak decoder norms at step 0, while emergent features reach their peaks between step 1000 (snapshot 10) and step 127000 (snapshot 30), with most peaking around step 6000 (snapshot 15). See Figures 3 and 4(a).
> 2. Initialization features correspond to token-level patterns (which is expected given their activation in the random initialization), whereas emergent features encode more complex information, including bigrams, phrases, and context-level patterns. See Figure 5 and Figure 24 for examples of emergent features.

---

> ### Author Response · Authors · 2025-11-21
> **Generalization across Model Families/Seeds (5/6)**
>
> > (Reviewer zNuD) How stable are the observed feature dynamics across different base models or seeds? Would larger or differently initialized models exhibit the same “turning point” behavior?
>
> > (Reviewer uCjx) Could you produce similar analysis with additional models with available checkpoints, such as Stanford CRFM GPT-2 or Olmo?
>
> We appreciate the reviewers' concerns regarding the generalizability of our methods and analyses across different model families and initializations. In addition to our original results on Pythia 160M and Pythia 6.9B (which demonstrate generalizability across model sizes), we conducted experiments on two additional Pythia 160M models with different random seeds (Pythia 160M Seed1 and Seed2) [1] and the Alias run of Stanford CRFM's GPT-2 model [2].
>
> - **Different initializations (Pythia 160M variants):** We observe highly consistent results across differently initialized versions of Pythia 160M. Decoder norms of initialization features follow identical trajectories, and emergent features begin rising around step 1000, exhibiting a clear two-phase pattern with the same transition point (Figure 25 and 26 of the revised manuscript). Attribution experiments confirm that we successfully identify sparse crosscoder features with significant contributions to downstream tasks. We show these results in Figure 28 and 29 of the revised manuscript.
>
> - **Different base model (GPT-2):** For Stanford CRFM's GPT-2 model, we observe a clear two-phase pattern, with emergent features rising around step 1,000. Upon manual inspection of these features, we still find that initialization features exhibit token-level information, while more complex patterns emerge in emergent features.
>
>   However, we find a notable difference: due to absolute positional embeddings, initialization feature norms do not peak at the beginning but rather at later steps (with above-threshold norm at beginning). This difference arises because the absolute positional embeddings initially exhibit large norms (larger than the word embeddings) and dominate the activation norm. Since we normalize all activation sources to the same average activation norm $\sqrt{d_\text{model}}$, this large positional contribution substantially reduces the observed norms of initialization features. We show these results in Figure 27 of the revised manuscript.
>
>   To sum up, the statistical-to-feature-learning phase transition still exists across models, despite small differences. Attribution experiments (Figure 30 of the revised manuscript) again confirm the effectiveness of sparse crosscoder features, with generally similar feature evolution patterns for specific downstream tasks.
>
> - **Other models with accessible checkpoints:** We also investigated other model families, including [BLOOM](https://arxiv.org/abs/2211.05100) and [OLMo](https://github.com/allenai/OLMo). Unfortunately, these models either lack sufficient intermediate checkpoints for fine-grained analysis of pretraining dynamics (BLOOM), or lack sufficiently early checkpoints to capture the two-stage transition (BLOOM and OLMo).
>
> We include all these findings above in Appendix G.1. We believe there is still room for more consolidated experiments supporting our claim. But due to limitations of existing resources for pretraining & scaling research, we are unable to go further in this direction for now. Nonetheless, our current experiments on 3 seeds of Pythia-160M (original Pythia suite + 2 PolyPythia seeds), 2 sizes of Pythia models (160M & 6.9B) and 2 model families (Pythia + stanford GPT2) can serve as strong evidence.
>
> [1] [PolyPythias: Stability and Outliers across Fifty Language Model Pre-Training Runs](https://arxiv.org/abs/2503.09543)
>
> [2] https://github.com/stanford-crfm/mistral

---

> ### Author Response · Authors · 2025-11-21
> **Generalization across Datasets (6/6)**
>
> > (Reviewer zNuD) Could the authors clarify the rationale behind choosing SlimPajama as the pretraining dataset? Were other datasets considered, and if so, why were they excluded?
>
> > (Reviewer QXzd) Have the authors verified that the same cross-coder features generalize across layers or different datasets?
>
> We appreciate the reviewers' concerns regarding the generalizability of our methods and analyses across different datasets. SlimPajama is a widely used deduplicated text corpus for pretraining that has been employed in prior work on sparse dictionary learning [1]. We selected SlimPajama primarily because it is a standard corpus with sufficient data volume for our experiments.
>
> To further demonstrate the robustness of our methods and analyses across datasets, we trained crosscoders for Pythia 160M on the FineWeb-Edu dataset [2], which consists of educational web pages. Our results show nearly identical patterns to those observed with Pythia 160M crosscoders trained on SlimPajama, exhibiting very similar feature evolution dynamics. We have incorporated detailed results from these experiments in Appendix G.2.
>
> [1] [Llama Scope: Extracting Millions of Features from Llama-3.1-8B with Sparse Autoencoders](https://arxiv.org/abs/2410.20526)
>
> [2] [The FineWeb Datasets: Decanting the Web for the Finest Text Data at Scale](https://arxiv.org/abs/2406.17557)

---

> > ### Comment · Reviewer_zNuD · 2025-11-28
> >
> > Thank you for the detailed response and for the additional experiments on more models. I appreciate the time that went into the rebuttal.
> >
> > * **Novelty.** I appreciate the clarification on the contribution regarding feature-level pretraining dynamics. At the same time, this clarification does not yet fully establish conceptual or methodological novelty; the work still appears as an extension of an existing tool. If the main innovation lies in analyzing pretraining dynamics, it would be helpful to explain why linear features extracted by crosscoders should faithfully summarize changes in model weights. In other words, can we quantify the gap between what the crosscoder lens captures and what is actually changing during training? In the end, by using crosscoders or any other probing-based methods, we are putting some “lens” into the model. While these lenses give us some signals, we do not know how *lossy* these signals are. For methods like crosscoders, they invent the “lens” for us to examine, so the novelty is methodological; for the observational work like this one, I would say the novelty lies in rigorous validation of whether this lens truly reflects the dynamics of interest and what aspects it may miss. For instance, would different interpretability lenses reveal similar patterns? Or *does the method itself shape what is observed*? From this perspective, many of the methodological details in Appendix A seem essential for understanding the tool. Their absence from the main text may unintentionally obscure important considerations for readers.
> >
> > * **Rationale for experimental metrics.** I understand the choice to use visualizations as a central form of evidence, and the visualizations in the paper are indeed strong. Still, visual clarity alone is not sufficient to support a general claim about model learning dynamics. The additional experiments on more models and datasets certainly improve the credibility of the findings, and I appreciate that extension. However, if the aim is to make an observational or analytic statement about consistent patterns across pretraining regimes, a broader and more systematic investigation would help lift the insights from “interesting observations” toward more genuine, stronger interpretability insights.
> >
> > * **Contextualizing n-gram learning in the literature.** The statement that “early training almost exclusively learns uni- and bi-gram distributions,” followed by a long block of citations, compresses several distinct strands of prior work. Expanding this discussion, as mentioned in your response, would clarify how your findings relate to each study. In particular, what does “almost exclusively” mean here? Does it imply that later training no longer develops uni-/bi-grams? This interpretation appears to contrast with prior findings (e.g., Nguyen 2024; Chen et al. 2024), which show that some complex bigrams continue to refine throughout training. Clarifying this point would help reconcile your conclusion with the broader literature and clarify its generality.
> >
> > * **Experimental choice.** Could you elaborate on why OLMo does not provide sufficiently early checkpoints for your analysis? OLMo seems like a natural fit, with dense checkpoints and the Dolma training dataset available. This is especially convenient when comparing model-predicted n-grams with the true n-grams, since infini-gram (Liu et al., 2024) provides n-grams statistics for Dolma and avoids the need to approximate distributions by sampling from SlimPajama.
> >
> > In summary, I am keeping my score. The work would benefit from deeper empirical grounding or theoretical analysis to clarify what the crosscoder lens measures and what it may miss, as well as from improvements in clarity (for example, integrating key methodological details from Appendix A into the main text). With these refinements, the paper could move from an ad hoc report toward a more scientific narrative. I believe the work would be well suited for workshop presentation and further iteration.

---

> ### Author Response · Authors · 2025-11-29
> **Response to Reviewer zNuD (1/2)**
>
> We are very grateful for your detailed response. We really appreciate the effort put into providing further advice.
>
>
>
> **Novelty**
>
> > If the main innovation lies in analyzing pretraining dynamics, it would be helpful to explain why linear features extracted by crosscoders should faithfully summarize changes in model weights. In other words, can we quantify the gap between what the crosscoder lens captures and what is actually changing during training? In the end, by using crosscoders or any other probing-based methods, we are putting some “lens” into the model. While these lenses give us some signals, we do not know how *lossy* these signals are.
>
> ​Our intended innovation indeed lies in pretraining dynamics analysis. It is true that quantifying the faithfulness of these methods would be a huge breakthrough in interpretability. And we actually designed a series of experiment to justify this. We introduce a straightforward intuition on why crosscoder decoder norms may faithfully indicate feature evolution, and detailed how we conduct a sanity check with linear probes in Appendix C. To clarify, though crosscoders (as a self-supervised method) and probes (supervised) both find linear directions sensitive to a specific concept, we believe they are two distinct methods since crosscoders are not explicitly trained to isolate any concept. And we did find a correlation of -0.867 between probing errors and crosscoder decoder norms. This is better understood as "when decoder norm of a feature firing on a specific concept goes up, probing loss on that concept goes down". In addition, using probing to evaluate the quality of sparse autoencoders has been adopted in existing literature ([SAEBench by Karvonen et al.](https://arxiv.org/pdf/2503.09532), [Scaling & Evaluating SAEs by Gao et al.](https://arxiv.org/abs/2406.04093)).
>
> ​We would like to also share more of our thoughts on how we try to balance generating interpretability hypotheses by reporting analysis results and scientific rigor in this work. Our crosscoders were only trained to reconstruct model activations under sparsity constraints without specifically trained on any given tasks. And we report in Section 5 that correlation is observed between decoder norm surge of several features related to plural nouns / postposition attributives and subject-verb agreement performance emergence. We also conducted a number of ablation studies to show downstream causality of these semantically related features. And more details are provided in Appendix E showing how this generalize to other 2 tasks along with additional details of our method. We currently cannot give a conjecture about how these results might fool us and we look forward to your take on this.
>
> ​Two potential failure modes of crosscoder feature alignment are discussed in Appendix A.6. And we report cases when these do happen in the wild.
>
> ​We reiterate many of our results above since they have not been included in the previous discussion period. We want to make sure you did not miss some of the points (if any).
>
> ​At the end of the day, we think the mechanistic faithfulness of sparse dictionary learning based methods remains an open question in this field. But based on the experimental results we provided in our initial manuscript and our extended analysis, we think there is enough evidence to mostly justify our findings.
>
> > for the observational work like this one, I would say the novelty lies in rigorous validation of whether this lens truly reflects the dynamics of interest and what aspects it may miss. For instance, would different interpretability lenses reveal similar patterns? Or *does the method itself shape what is observed*?
>
> ​	As for the two possibilities you mentioned, they actually touch the heart of this field. But we think we have resolved to an acceptable extent in the above discussion, i.e., probing results & causality of these features.
>
> ​	We are very grateful for your suggestions on how we should refine our work. However, we think connecting microscopic findings with macroscopic model behaviors are also exciting findings, i.e. our two-stage feature emergence and connection with downstream performance on induction, IOI and SVA tasks. We respect your opinions on this, but we do believe in the novelty of these findings.
>
> > From this perspective, many of the methodological details in Appendix A seem essential for understanding the tool. Their absence from the main text may unintentionally obscure important considerations for readers.
>
> Could you be more specific about which part should be added to the main text?

---

> ### Author Response · Authors · 2025-11-29
> **Response to Reviewer zNuD (2/2)**
>
> **Rationale for experimental metrics**.
>
> We thank you for your acknowledgement on credibility improvement.
>
> > However, if the aim is to make an observational or analytic statement about consistent patterns across pretraining regimes, a broader and more systematic investigation would help lift the insights from “interesting observations” toward more genuine, stronger interpretability insights.
>
> ​Could you provide more actionable feedback on what kind of broader and more systematic investigation you think would be helpful? And which part of the current form of the manuscript fail to meet the standard of "genuine, stronger interpretability insights"?
>
> **Contextualizing n-gram learning in the literature.**
>
> > In particular, what does “almost exclusively” mean here? Does it imply that later training no longer develops uni-/bi-grams? This interpretation appears to contrast with prior findings (e.g., Nguyen 2024; Chen et al. 2024), which show that some complex bigrams continue to refine throughout training. Clarifying this point would help reconcile your conclusion with the broader literature and clarify its generality.
>
> ​We quote the rationale for this statement provided in Section 6: "the training losses during this period approach the theoretical minimum achievable if the model perfectly matched the true token distributions—i.e., the entropy of these distributions."
>
> ​This does not imply "later training no longer develops uni-/bi-grams", as the word "almost" suggests. Thus this do not contrast with prior findings.
>
> **Experimental choice**.
>
> > Could you elaborate on why OLMo does not provide sufficiently early checkpoints for your analysis? OLMo seems like a natural fit, with dense checkpoints and the Dolma training dataset available.
>
> Our experiments show that the phase changes we observed in Pythia models and GPT2 models are both at ~1k steps, with batch sizes as follows:
>
> - Pythia: $32 \times 32 \times 2048 = 2097152$
> - GPT2: $16 \times 8 \times 1024 = 131072$
> - OLMo: $128 \times 4 \times 4096 = 2097152$
>
> Therefore, we suspect access to checkpoints earlier than 1k step for OLMo models is required to test generalization of our findings. In brief, OLMo checkpoints are dense, but not at the logarithmic scale.
>
> In summary, we respect your unwillingness to update your score. But we think we have resolved many of your raised concerns. And further improvement might require clearer and more actionable feedback.

---

### Author Response · Authors · 2025-12-03
**Summary of Main Contributions and Reviewer Comments (1/3)**

**Dear PCs, SACs, ACs, and Reviewers,**

We would like to express our sincere gratitude to the PCs, SACs, ACs, and Reviewers zNuD, uCjx, QXzd, and effr for their valuable time, constructive feedback, and thoughtful discussions during the review process. We provide below a concise summary of our paper’s main contributions, along with a synthesis of the reviewers’ comments and the key points raised during the discussion.

------

- This paper enables to **track feature evolution** across LM pre-training snapshots. Through **crosscoders**, we can observe where features emerge, rotate, and degenerate, thereby providing novel lens into the **internal dynamics of model pre-training**.
  - All the 4 reviewers acknowledge the **importance of our research question** (zNuD: Strength 1, 2, 3; uCjx: Strengths; QXzd: Strengths; effr: Strength 1) and its alignment with the **current interest of mechanistic interpretability field** (effr: Strength 1).
  - Regarding the **validity of decoder norms as proxies of feature strength** (QXzd: Question 2; effr: Weakness 1), we clarified the sanity check we had performed with **linear probes** where a **correlation of -0.867** exists between probing errors and crosscoder decoder norms.

- We perform in-depth analyses on cross-snapshot features, showcasing both **general and per-feature evolutionary properties**. We further connect **microscopic features to macroscopic downstream model behaviors**.
  - Our analyses on feature emergence and causal studies on feature-to-behavior connection were highly acknowledged by reviewers (uCjx: Strengths; QXzd: Strengths; effr: Strength 2).
  - Regarding the **scale of x-axes in our figures** (uCjx: Weakness 1 & Question 1; effr: Weakness 2 & Question 1), we clarified the **exact snapshot selection** in our main experiments and **how x-axes scale with it**. We further provided detailed explanations on **why this near-exponential sampling is reasonable and essential** for our demonstration.
  - Regarding the **automated interpretation & complexity score evaluation via Claude Sonnet 4** (effr: Weakness 3), we provided a literature review of existing usage of LLMs in feature interpretation with both positive and negative evidences. We claimed that the automated interpretation does **provide a labor-free method for rapid feature evaluation** based on existing literature and **our manual verification of the LLM results**.
  - Regarding the **validity of attribution patching** (QXzd: Question 3; uCjx: Question 1), we provided a list of literature presenting and verifying the causal validity of attribution patching. We additionally included **random baselines** in our attribution results, showing ablating random features hardly affect the original model behavior.

---

> ### Author Response · Authors · 2025-12-03
> **Summary of Main Contributions and Reviewer Comments (2/3)**
>
> - We provide evidence for the phase transition from **a statistical learning phase to a feature learning phase**.
>   - Regarding the **definition of initialization features and emergent features** (zNuD: Weakness 4 & Question 3), we established a **straightforward symmetrical formal definition** for these feature groups, and further clarified distinct properties of these features, which make them stand out.
>   - Regarding the **generalizability of our method, analyses and results** **across models** (zNuD: Weakness 1; uCjx: Question 2; effr: further comment), **across datasets** (zNuD: Weakness 1 & Question 1; QXzd: Question 4), **across layers** (QXzd: Question 4), and **across spacing of snapshot selections** (QXzd: Question 1), we additionally performed comprehensive experiments across various dimensions of settings.
>     - We conducted experiments on two additional Pythia 160M models with different random seeds and one equally sized Stanford CRFM's GPT-2 model, observing highly consistent results where all models come with clear two-phase patterns and well connections with downstream tasks. We provided theoretical explanations on the evolutionary distinctions of GPT-2 and Pythia models, as well as why other model families might not well serve as good analyzing target. Nonetheless, our current experiments on **3 seeds of Pythia-160M (original Pythia suite + 2 PolyPythia seeds), 2 sizes of Pythia models (160M & 6.9B) and 2 model families (Pythia + stanford GPT2)** can serve as strong evidence.
>     - We explained rationale behind our choosing SlimPajama as the pretraining dataset. We further conducted experiments on the **FineWeb-Edu dataset** to demonstrate the robustness of our methods and analyses across datasets, which the results are basically the same.
>     - We clarified our existing results on **all layers of Pythia 160M**.
>     - We conducted additional experiments on **different numbers of pretraining snapshots** downsampled from the original set, which the results show nearly identical trend with only a reduction in temporal resolution.
>     - All these additional experiments are incorporated in the revised manuscript.
>   - Regarding **previous results of statical phases** (zNuD: Weakness 3 & Question 4), we contextualized more n-gram related literature in our manuscripts, while providing brief summaries of existing cited research on models learning statistical patterns, which we think might be more directly related to our findings. Moreover, we provided detailed clarification and comparisons in how our presenting results of the statistical phases are consistent with previous research.

---

> > ### Author Response · Authors · 2025-12-03
> > **Summary of Main Contributions and Reviewer Comments (3/3)**
> >
> > - Regarding the remaining concerns of **how learning rate schedules might cause features to emerge** (Reviewer effr) and the **novelty** of our methodology (Reviewer zNuD), we would like to offer a unified clarification (which is already presented in the rebuttal):
> >   - Our intended key novelty lies in **our crosscoder-based analyses specific to pre-training dynamics**. These analyses include the causal connection of microscopic features to macroscopic behavioral changes, the relevance of feature complexities and emergence timing, and the detection of the emergence of superposition. Besides, our technical improvements on crosscoder training ensures the feasibility of crosscoder lens, including the architectural designs and parallism of training.
> >   - Reviewer zNuD **assumed our novelty should lie in rigorous validation of how lossy signals provided by crosscoders are**. To this end, we had introduced the sanity check through linear probes (as previously mentioned) that ensure the faithfulness of our crosscoder lens. Nevertheless, we think the scope of this work is **beyond** simple validation of *"crosscoders can work"*, but involving further analyses on the results of the lens, showing ***"how crosscoders reveal interesting microscopic properties of training dynamics"***. We would like to emphasize that our presenting methods of causal connection and superposition detection is neglected to some extent.
> >   - Reviewer effr **concerned that the two distinct learning phases could be explained by "a large enough learning rate makes feature begin to emerge"**, which he thought to be trivial. Despite our validation across different model initializations and different base models, we could not rule out the assumption due to the unconventionality of training without an LR warmup. However, we think this does not harm our effectiveness of our findings. **Our crosscoder lens should be more like a microscope on pretraining dynamics. The phenomenon revealed by the microscope may itself have various origins, but as our uncovered phenomenon is highly consistent with previous studies, we demonstrate the effectiveness of our microscope**. We have detailed **how two phases emerge** in our paper, and we think the exploration of the underlying mechanism of **why two phases emerge** is currently beyond our scope.
> >
> > We have faithfully summarized the contributions of our work, the reviewers’ comments, and our corresponding responses above, and we hope this will assist the ACs in their assessment. We sincerely thank the reviewers, ACs, SACs, and PCs for their hard work and outstanding efforts. Their insightful feedback has further improved the quality of our paper.
> >
> > Best regards,
> >
> > The Authors

---

### Meta-Review · Area_Chair_kDvf · 2025-12-29

**Summary:**

This paper investigates the microscopic evolution of internal features during LLM pre-training using Crosscoders. The authors identify a distinct phase transition from a "statistical learning phase" (n-gram fitting) to a "feature learning phase" (sparse concept emergence) and establish causal links between feature emergence and downstream task performance.

While the initial reviews were mixed (Scores: 2, 4, 6, 8), the authors provided an exceptionally strong rebuttal. They expanded experiments across different random seeds, model architectures (GPT-2), and datasets (FineWeb-Edu) to prove robustness. The decision to Accept (Poster) is based on the significant empirical value of "opening the black box" of training dynamics. The novel application of Crosscoders to the temporal dimension provides high-resolution insights that outweigh the remaining concerns about theoretical purity.

**Reviewer Concerns:**

Concerns Addressed:

- Generalizability: Reviewers (zNuD, QXzd) questioned if findings were specific to Pythia/SlimPajama. The authors successfully addressed this by conducting extensive additional experiments on GPT-2 and the FineWeb-Edu dataset, showing consistent two-phase dynamics.

- Validity of Metrics: Concerns about whether Decoder Norms are a valid proxy for feature strength were addressed by demonstrating a strong correlation (-0.867) with linear probing errors (Appendix C).

- Definitions: The authors provided formal definitions for "initialization" and "emergent" features as requested.


Outstanding Concerns (and why they are overruled):

- Tool Faithfulness (Reviewer zNuD): The reviewer argues that Crosscoders act as a "lens" that might distort reality and demands quantification of this gap. While valid, this is an open problem in the entire field of mechanistic interpretability and sets an unreasonably high bar for an empirical observation paper.

- Confounding Factors / Triviality (Reviewer effr): The reviewer suspects the phase transition is a trivial artifact of learning rate warmup. However, the authors showed that the transition point (approx. step 1000) does not align with the warmup schedule (step 1430) and persists across models with different schedules. I find the authors' empirical evidence more convincing than the reviewer's skepticism.

**Reviewer Scores:**

Reviewer zNuD (Current: 2): 2. This reviewer explicitly stated they would keep their score, remaining unconvinced by the novelty despite the rebuttal.

Reviewer uCjx (Current: 8): 8. This reviewer was highly positive about the novelty and analysis from the start.

Reviewer QXzd (Current: 6): 6/8. This reviewer was borderline but positive. Given the successful addressal of generalization concerns (their main question), they would likely view the paper more favorably.

Reviewer effr (Current: 4): 4. This reviewer acknowledged the extra experiments but maintained their score due to a desire for a stronger mathematical framework, which is arguably out of scope for this empirical study.

---

### Decision · Program_Chairs · 2026-01-26

Accept (Poster)